# A transcriptional switch controls sex determination in *Plasmodium falciparum*

A. R. Gomes[1], A. Marin-Menendez[2,4], S. H. Adjalley[3,4], C. Bardy[2,4], C. Cassan[2,4], M. C. S. Lee[3] & A. M. Talman[2✉]

Sexual reproduction and meiotic sex are deeply rooted in the eukaryotic tree of life, but mechanisms determining sex or mating types are extremely varied and are only well characterized in a few model organisms[1]. In malaria parasites, sexual reproduction coincides with transmission to the vector host. Sex determination is non-genetic, with each haploid parasite capable of producing either a male or a female gametocyte in the human host[2]. The hierarchy of events and molecular mechanisms that trigger sex determination and maintenance of sexual identity are yet to be elucidated. Here we show that the male development 1 (*md1*) gene is both necessary and sufficient for male fate determination in the human malaria parasite *Plasmodium falciparum*. We show that Md1 has a dual function stemming from two separate domains: in sex determination through its N terminus and in male development from its conserved C-terminal LOTUS/OST-HTH domain. We further identify a bistable switch at the *md1* locus, which is coupled with sex determination and ensures that the male-determining gene is not expressed in the female lineage. We describe one of only a few known non-genetic mechanisms of sex determination in a eukaryote and highlight Md1 as a potential target for interventions that block malaria transmission.

Sexual differentiation in *Plasmodium* starts when intraerythrocytic parasites become committed to a sexual fate under the control of the transcription factor AP2-G[3–6]. Sexually committed progenitor parasites undergo a secondary cell fate decision to become either a male or a female gametocyte (Fig. 1a)[7]. A recent genetic screen identified several genes with a role in male and female development in the rodent model *Plasmodium berghei*, including Pb *md1* (ref. [7]). However, the molecular basis of sex determination remains unknown.

## *md1* is necessary to determine male fate

To investigate the role of *md1* (PF3D7_1438800) in the human parasite *P. falciparum*, we generated a knockout (KO) line by using CRISPR–Cas9 (Extended Data Fig. 1a,b). KO parasites had no asexual growth defects (Extended Data Fig. 1c) and produced gametocytes that matured as the wild type (WT) (Extended Data Fig. 1d) with no loss in viability (Extended Data Fig. 1e). We quantified the relative sex ratio by using an immunofluorescence assay (IFA) of the female-specific marker Pfg377 (ref. [8]). WT gametocytes were typically female-biased, with 85% Pfg377-positive cells (Fig. 1b). By contrast, KO gametocytes were almost exclusively Pfg377-positive (97%), indicating few, if any, males in the population. Concordantly, the relative abundance of the male marker PfMGET was dramatically reduced[9] (Extended Data Fig. 1f), and exflagellation (the hallmark of male gametogenesis) was not observed in the KO lines (Extended Data Fig. 1g). Episomal re-introduction of full-length (FL) *md1* (KO compl.) restored the sex ratio to WT levels, confirming that *md1* disruption drives the loss of males (Fig. 1b and Extended Data Fig. 1f).

To understand the nature of this phenotype, we pooled gametocytes at days 3, 5, 7 and 9 post-stress for both the WT and the KO line, and characterized these pools by using droplet single-cell transcriptomics (10x). We confirmed the capture of both male and female lineages by using sex-specific markers (Fig. 1c) and by mapping onto the reference malaria cell atlas[10,11] (Extended Data Fig. 1h). Next, a trajectory-inference analysis assigned each cell to the sexual progenitor, the male or the female states, and ordered the cells along two developmental paths (Fig. 1c), with very few contaminating asexual parasites (Fig. 1c and Extended Data Fig. 1h). A comparison of the trajectories revealed that the WT-produced cells were present in both male and female lineages. In contrast, the KO line had a completely altered topology with nearly complete absence of cells assigned to the male branch (Fig. 1d,e). Although a few cells (4 out of 2,681) were still assigned to the male lineage in the KO, they corresponded to the very first pseudotime points at the base of the male branch and, therefore, were probably still in the progenitor state (Fig. 1e). We conducted a differential expression analysis between the cells immediately preceding and following the sex-determining event and identified genes that were upregulated or downregulated during the earliest steps of male differentiation (Fig. 1f). We then compared the expression of these genes in cells assigned to the progenitor or male state for both WT and KO cells (Fig. 1f). We found that predicted KO male cells still displayed a progenitor signature, indicating that determination to a male fate does not occur in the KO line. In addition, we did not detect arrested and/or dead or any other cell populations in the KO line that were not present in the WT (Fig. 1d and Extended Data Fig. 1i), confirming a determination rather than a developmental defect.

[1]Laboratory of Pathogen Host Interactions UMR 5235, Université de Montpellier and CNRS, Montpellier, France. [2]MIVEGEC, Université de Montpellier, IRD, CNRS, Montpellier, France. [3]Wellcome Sanger Institute, Hinxton, UK. [4]These authors contributed equally: A. Marin-Menendez, S. H. Adjalley, C. Bardy, C. Cassan. ✉e-mail: arthur.talman@ird.fr

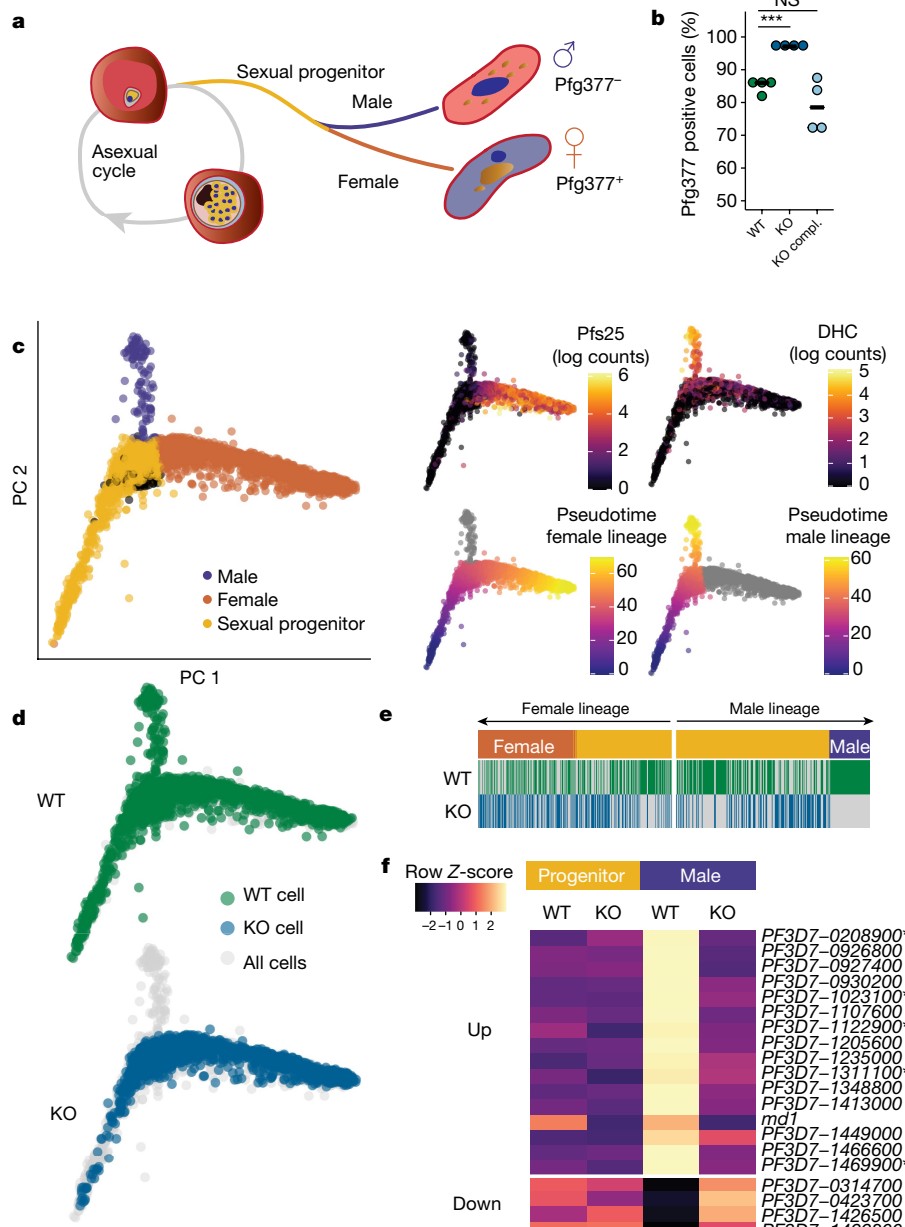

**Fig. 1 | *md1* is necessary to determine a male fate. a**, Asexually replicating parasites can bifurcate to sexual development, which is initially characterized by a sexual progenitor state from which the parasite subsequently differentiates into either male or female. Males and females have distinct gene and protein expression programmes; the Pfg377 protein used in this study is exclusively found in females[8]. **b**, Proportion of Pfg377-positive gametocytes by immunofluorescence in WT, KO and a complemented KO (KO compl.) (four biological replicates) (two-sided *t*-test: WT versus KO, *P* = 0.0006711; two-sided *t*-test: WT versus KO complemented, *P* = 0.182). **c**, Combined principal component (PC) analysis of single-cell transcriptomes covering bifurcation of sexual progenitor cells into male and female branches as identified by female (Pfs25) and male (DHC, PF3D7_0905300) sex-specific markers[17]. Contaminating asexuals are black in the left panel. Cells are also ordered along two developmentally distinct pseudotimes (lower right panels). **d**, WT and KO cells are highlighted. **e**, Representation of either genotype along each pseudotime path, with each cell being represented by a green (WT) or blue (KO) bar placed along either of the developmental paths. Although WT and KO cells are represented throughout the female lineage, hardly any KO cells are mapped to the male lineage. **f**, To identify the first genes in the male developmental path, 50 cells on either side of the male sex-determining event were used to conduct a differential expression analysis. Expression of these genes are displayed for each genotype just before (progenitor, 50 WT versus 50 KO) and just after the sex-determining event (male, 50 WT versus 4 KO cells), showing that the KO fails to activate and/or repress the expression of genes at the earliest points of male differentiation. Genes important in male gametocyte development are denoted by an asterisk.

Altogether these data indicate that *md1* disruption prevents bifurcation to a male fate and establishes *md1* as necessary for male determination.

## *md1* is sufficient to determine male fate

*md1* is only found in the Haemosporida and Piroplasma orders of the Apicomplexa[12] (Extended Data Fig. 2a). In *P. falciparum*, Md1 contains two conserved C-terminal domains: an OST-HTH/LOTUS domain and an OST-HTH-associated domain (OHA) (Extended Data Fig. 2b). Md1 also contains an N-terminal extension (NTE) with no conserved homology or predicted domains (Extended Data Fig. 2b). Interestingly, an early KO version that partially disrupted the *md1* locus (Δ270–699), leaving the NTE largely intact (Extended Data Fig. 2c,d), showed a marked increase in the relative abundance of a male marker by quantitative

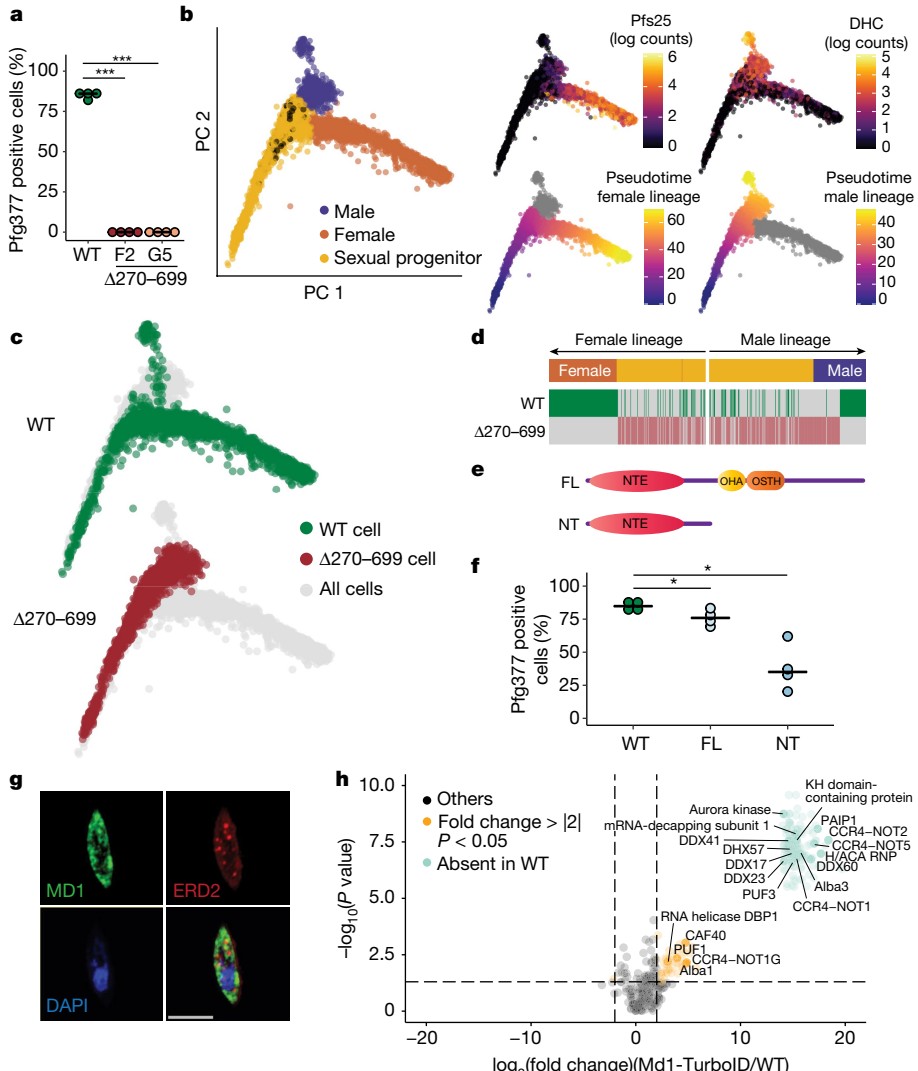

**Fig. 2 | *md1* is sufficient to determine a male fate. a**, Proportion of Pfg377-positive gametocytes (female-specific) by IFA in WT and two clones of Δ270–699 (four biological replicates) (two-sided *t*-test: WT versus Δ270–699(F2), $P = 2.2 \times 10^{-16}$; two-sided *t*-test: WT versus Δ270–699(G5), $P = 2.2 \times 10^{-16}$). **b**, Combined principal component analysis of single-cell transcriptomes covering the bifurcation of progenitor cells into male and female branches as identified by female (Pfs25) and male (DHC, PF3D7_0905300) sex-specific markers[17]. Contaminating asexuals are black in the left panel. Cells are also ordered along two developmentally distinct pseudotimes (lower right panels). **c**, WT and Δ270–699 cells are highlighted. **d**, Representation of either genotype along each developmental pseudotime ordering. Each cell is represented by a green (WT) or brick coloured (Δ270–699) bar placed along either of the developmental paths. Although WT cells cover female and male lineages, no female cells were found in Δ270–699 and all male cells were arrested early in development. **e**, Schematics of Md1 protein versions translated from episomes. **f**, The proportion of Pfg377-positive gametocytes in over expression lines (four biological replicates) (two-sided *t*-test: WT versus FL, $P = 0.04122$; two-sided *t*-test: WT versus NT, $P = 0.01104$). **g**, The IFA of an Md1-3xHA gametocyte showing Md1 is present in cytoplasmic foci and absent from the nucleus (4′,6-diamidino-2-phenylindole (DAPI)) or Golgi (ERD2); scale bar, 5 μm. This representative observation was made in three different independent experiments. **h**, A volcano plot of proteins identified by proximity labelling followed by MS of Md1-TurboID versus WT (untagged). Notable orthologues of proteins associated with RNP granules in *Plasmodium* or predicted interactors of proteins containing LOTUS domains in other systems[24] are highlighted and listed in Supplementary Table 2. The *t*-test used was two-sided and was corrected for multiple testing with the false discovery rate (FDR) method.

reverse transcription–PCR (qRT–PCR) (Extended Data Fig. 2e). In fact, we did not observe a single female in either of two Δ270–699 clones by IFA (Fig. 2a). However, despite no apparent loss in viability (Extended Data Fig. 2f), these male gametocytes failed to exflagellate (Extended Data Fig. 2g) and displayed an aberrant morphology (Extended Data Fig. 2h,i), suggesting a developmental arrest. To understand the Δ270–699 phenotype, we generated a droplet single-cell transcriptomic data set covering gametocyte maturation (Fig. 2b and Extended Data Fig. 3a). By using the same trajectory-inference approach as above, we observed a complete absence of the female lineage in Δ270–699 with all parasites after the progenitor state mapping to the early male

lineage (Fig. 2c,d). We saw no increase in poor-quality cells that would indicate cell death and/or arrest (Extended Data Fig. 3b). Importantly, a truncated form of *md1* was transcribed (Extended Data Fig. 3c,d) and translated (Extended Data Fig. 3e), suggesting that the truncated protein is sufficient to determine male fate and prevent bifurcation to the female branch. We reasoned that male determination requires the NTE but not the OST-HTH and OHA domains, which are absent in Δ270–699. We, therefore, introduced episomes encoding either the NTE portion (N terminus, NT) or FL *md1* under the control of the *md1* promoter in a WT background (Fig. 2e). The proportion of Pfg377-positive cells was reduced to 76% in FL and 38% in NT versus 85% in WT (Fig. 2f).

Relative qRT–PCR showed an increase in the number of males in the NT parasites (Extended Data Fig. 3f), confirming the male-determining function of the Md1 N terminus. The developmental arrest seen in Δ270–699 males (Fig. 2c,d) was suggestive of a developmental role for the OST-HTH and OHA domains. Indeed, Δ270–699 male cells failed to appropriately regulate a number of genes important in male differentiation (Extended Data Fig. 3g), indicating a derailed transcriptional programme in the absence of the C terminus of Md1 during male gametocytogenesis.

We next sought to explore the function of Md1 by using a C-terminal HA-tagged parasite line (Md1-3xHA) (Extended Data Fig. 4a–c). Interestingly, Md1 localized to cytoplasmic foci and not the nucleus (Fig. 2g and Extended Data Fig. 4d), suggesting that it is not a transcription factor. We next conducted proximity labelling by fusing biotin ligase to the C terminus of Md1 (Md1-TurboID, Extended Data Fig. 4e–g), followed by the identification of biotinylated proteins by mass spectrometry (MS) and enrichment analysis relative to an untagged WT control. Biotinylated proteins in the Md1-TurboID strain included known factors involved in translational repression in *Plasmodium* (Fig. 2h and Supplementary Table 2), such as members of the Alba[13] and PUF[14] families, and most elements of the CCR4–NOT complex, including the *Plasmodium* specific paralogue CCR4–NOT1G, which has been linked to male fertility in the rodent model *Plasmodium yoelii*[15]. We also identified orthologues of yeast and metazoan ribonucleic granule proteins (RNP) such as dead-box RNA helicases (DDX17, DDX23, DDX41 and DDX60), KH-domain-containing proteins and messenger RNA (mRNA) decapping enzymes. Interestingly, we also detected orthologues of TDRD7 interactors, an OST-HTH-domain-containing protein involved in spermatogenesis in metazoans[16], such as the PAIP1 and DExH-Box helicases (Fig. 2h, Extended Data Fig. 4h and Supplementary Table 2). Our cellular analysis suggests that Md1 associates with cytoplasmic RNP granule-like structures. Further analysis of these structures and their associated RNAs may reveal how these granules govern and coordinate the determination to a male fate and the development of male gametocytes.

## A sex-determining transcriptional switch

Analysis of a bulk RNA-seq dataset identified transcription at the *md1* locus in both sense and antisense (AS) orientations[17]. To characterize the RNA species transcribed from the *md1* locus, we examined the RNA content of cells undergoing sexual differentiation by direct RNA long-read sequencing. Three main RNA species were transcribed from the *md1* locus: (1) FL mRNA (exons 1–4; *md1*-mRNA); (2) AS RNAs transcribed from within intron 1 of the annotated *md1* gene; and (3) a shorter sense RNA with only exons 2–4 (*md1*-shortRNA) (Extended Data Fig. 5a). Multiple AS isoforms of approximately 2.5–2.9 kb were present with several introns located in the annotated *md1* exon 1 region (Extended Data Fig. 5a). Splicing patterns were confirmed by analysing split reads in single-cell RNA-seq data capturing exon–exon junctions, which can only stem from a single orientation (Extended Data Fig. 5b). The coverage pattern was suggestive of this 2.5–2.9 kb form being the main AS species with shorter fragments probably arising from partially sheared and/or degraded RNA, as has been reported for direct RNA-seq data[18]. As this long AS RNA lacks coding potential (coding probability = 0.000148)[19], we refer to it as *md1* long non-coding RNA (*md1*-lncRNA) (Extended Data Fig. 5c). As coverage suggested that the transcription start site of the *md1*-lncRNA spanned intron 1 of the *md1* gene (Extended Data Fig. 5a), we removed intron 1 leaving the locus otherwise intact by using CRISPR–Cas9 (Extended Data Fig. 5d,e) and named this line Δint1. We saw an approximately 5 log fold reduction of *md1*-lncRNA expression in Δint1 by using an AS intron-spanning qRT–PCR (Extended Data Fig. 5c,f), confirming that the intron is indeed important for *md1*-lncRNA transcription. We further validated normal expression of sense transcripts by qRT–PCR (Extended Data Fig. 5c,g,h) and Md1 protein expression by shotgun proteomics (Extended Data

Fig. 5i). The Δint1 line displayed normal sexual maturation (Extended Data Fig. 5j) and exflagellation (Extended Data Fig. 5k). The Pfg377 IFA revealed a subtle, but notable, increase in the number of males in the Δint1 line (Extended Data Fig. 5l). Taken together, this suggests that *md1*-lncRNA is dispensable for male determination and its knockdown may even enhance bifurcation to a male fate.

To further understand the pattern of transcription and translation at the *md1* locus during sex determination, we integrated a C-terminal translational reporter in the endogenous locus, composed of a self-cleaving peptide (T2A) followed by green fluorescent protein (GFP) (Extended Data Fig. 6a,b). The Md1-2A-GFP clone was then sampled by cell sorting throughout sexual development and was analysed with full-length single-cell RNA-seq (Smartseq2)[20]. We used this alternative method as it allows us to preserve cellular information such as GFP intensity that is measured during the cell sort, and associate it with each transcriptome. By using the same trajectory-inference approach as above, we confirmed sampling over the sex-determining event and ordered cells in both male and female developmental paths (Fig. 3a and Extended Data Fig. 6c). The *md1* read counts (non-stranded) were first observed to be lowly expressed in sexual progenitors and were subsequently present in both female and male lineages (Extended Data Fig. 6c). We isolated reads spanning exon–exon junctions to map the directionality of transcription at the *md1* locus, and observed that expression of the exon 1–2 junction (which can only arise from *md1*-mRNA) was exclusively expressed immediately following bifurcation into the male branch (Fig. 3b,c). By contrast, AS transcription (the sum of all AS junctions detecting *md1*-lncRNA) was uniquely detected immediately following bifurcation into the female branch (Fig. 3b,c). A Fisher's exact test indicated that the expression of *md1*-mRNA and *md1*-lncRNA were expressed in a mutually exclusive fashion (odds ratio = 0.028, $P = 6.03 \times 10^{-10}$). We then mapped expression of the Md1 protein, as measured by the normalized GFP intensities recorded during the cell sort, onto the dataset and found that it was associated with bifurcation into the male lineage (Fig. 3b,c). Altogether these data indicate that there is a bistable switch at the *md1* locus (Fig. 3d). The *md1* locus first becomes active before cells can be identified as belonging to one sex or the other, although the *md1* RNA species cannot yet be identified. In the male lineage, *md1*-mRNA expression is seen as soon as males are unambiguously identifiable on a transcriptional level, and this is coupled with robust protein expression (Fig. 3b,c). Sense transcription and translation are seen throughout male maturation (Fig. 3b,c). In the female lineage, the activity of a second promoter, which spans intron 1 and produces the *md1*-shortRNA and the *md1*-lncRNA, is concurrent with the identification of cells as females on a transcriptional level and is maintained throughout development. Activity from the *md1* promoter is not observed in the female lineage (Fig. 3b,c).

We next assessed the stability of the switch when the locus is perturbed. The initial timing of activation of the switch during sex determination was not altered in the KO line or the Δ270–699 (Extended Data Fig. 6d), indicating the gene body is dispensable for the regulation of the locus in progenitors. We further postulated that the exclusive representation of males and truncated *md1* transcription in Δ270–699 (Extended Data Figs. 3c and 6d) could be attributed to the inability to silence sense expression from the main promoter. This implies that the rest of the gene body is necessary for silencing the male promoter in females. The subtle phenotype in Δint1 (that is, the slight increase of males; Extended Data Fig. 5l) indicates that *md1*-lncRNA is not the main driver of transcriptional silencing of the *md1* promoter in females. The two transcriptional states could, therefore, be the result of differential chromosome conformation or compartmentalization, or mutually exclusive binding of specific regulators either to the promoter or to intron 1 through exon 4. We next targeted a catalytically inactive Cas9 (dCas9) to the promoter region of *md1* by using a tiling array of single-guide RNAs (sgRNAs) (Md1-dCas9-prom; Extended Data Fig. 6e). We observed a shift of the sex ratio towards female in the Md1-dCas9-prom parasites

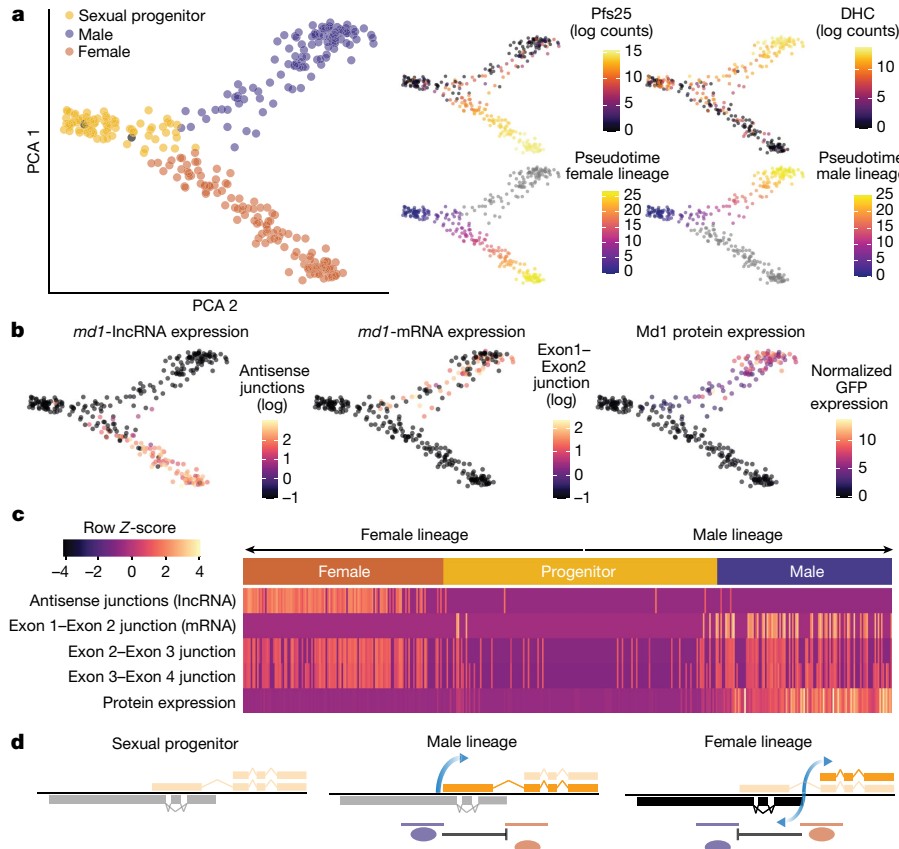

**Fig. 3 | Transcriptional and translational patterns at the *md1* locus are associated with the sex-determining event. a**, Principal component analysis of single-cell transcriptomes covering bifurcation of progenitor cells into male and female branches as identified by female (Pfs25) and male (DHC, PF3D7_0905300) sex-specific markers[17]. Contaminating asexuals are black in the left panel. Cells are also ordered along two developmentally distinct pseudotimes (lower right panels). **b**,**c**, Expression of different *md1* transcripts, as measured by exon–exon junction counts and normalized GFP intensity (Md1 protein) during the sex-determining event (**b**) and displayed along each developmental pseudotime ordering (**c**). **d**, Model of the three cellular states and associated *md1* locus during sex determination. Faded colours of the RNA species represent the absence of expression. Progenitors show no expression as there is a probable lag between activation in either state at the *md1* locus and transcriptional consequence (already committed but which cannot yet be sexed by single-cell transcriptomics). Male and female transcriptional states are displayed with their transcription pattern from the main promoter and the alternative bidirectional promoter, respectively. The proposed binding site for possible transcriptional activators in the male (purple line) and female (brown line) lineages are highlighted.

compared to Md1-dCas9-ctrl (without sgRNAs) (Extended Data Fig. 6f), suggesting that regulator occupancy and/or competition at the locus may be key in regulating mutually exclusive expression at the *md1* locus (Fig. 3d). Epigenomics and single locus proteomics may determine how this intricate regulation functions in future studies.

Commitment to sex and sex determination are seemingly separate cell fate decisions. Sexual commitment is thought to occur either in the preceding asexual cycle and give rise to an exclusively sexual progeny, or within the same cycle[21]. Earlier work suggested that a single committed schizont would mostly give rise to either male or female progeny, suggesting sex determination in the preceding asexual cycle[22,23], with the caveat that one of these studies observed some mixed progeny[23]. These studies, however, precede the discovery of same-cycle commitment for which the point of determination remains unexplored. Our observation that a transcriptional sexual progenitor state exists, which is neither male nor female and precedes determination, prompted us to evaluate the timing of transcriptional sex determination. Correlation to a bulk timecourse suggests gametocytes become transcriptionally dimorphic between stage II and III (Extended Data Fig. 6g). We further find that Md1-2A-GFP protein expression, which is coupled with the sex determination event (Fig. 3c), first appears around day 4 of gametocytogenesis (Extended Data Fig. 6h) and GFP-positive cells are stage III and later (Extended Data Fig. 6i). Although we do not discount the possibility that there is a transcriptionally silent sex-determined state, our data strongly suggest that sex determination occurs around day 4 of gametocytogenesis during the transition from stage II to III.

In summary, we have identified Md1 as a sex-determining effector in malaria parasites. Interestingly, we have found that only the N terminus of the protein was required to specify a male fate. This portion of the protein is not conserved outside hematozoa (Extended Data Fig. 2a), consistent with the notion that sex-determining genes are generally fast evolving[1]. By contrast, the Md1 C terminus contains an OST-HTH/LOTUS domain that is dispensable for sex determination but is important as a developmental effector in male gametocytes. Whether the two functions are exerted through the same molecular pathways, or whether this domain fusion is a safe-lock mechanism to ensure proper maturation after specification of a male fate remains unknown. Md1 localizes to cytoplasmic granules and interacts with RNP granule proteins, further highlighting LOTUS-containing proteins as hubs for developmental processes throughout the eukaryotic tree of life. Finally, the intricate gene regulation pattern at the *md1* locus provides further evidence for its centrality in sex determination. In genetically determined systems, cell fate control can rely on the presence or absence of sex-determining loci to orchestrate sex specification. We show that sex determination in genetically identical cells requires a robust regulatory architecture around the regulators of a cell's sexual fate.

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

## Methods

### In vitro parasites culture and phenotyping

Parasites (NF54) were cultured in O⁻ or A⁻ whole blood obtained from the Etablissement Français du sang (21PLER2020-004), in RPMI 1640 culture medium (Gibco) supplemented with 2 g l⁻¹ glucose (Sigma), 2 g l⁻¹ sodium bicarbonate (Sigma), 10 mg l⁻¹ hypoxanthine (Sigma), 50 mg ml⁻¹ gentamicin (Gibco), 0.5% Albumax-II (Gibco) and 5% AB+ human serum (21PLER2020-004), and were kept at 37 °C in a gaseous environment containing 5% $O_2$, 5% $CO_2$ and 90% $N_2$. Sexual stages were obtained by using the same complete media and with standard gametocyte culturing methods[25]. Contaminating asexual forms were removed from the cultures by treatment with 20 U ml⁻¹ heparin sodium salt (Sigma) added to the medium[26]. For the time courses used in the single-cell RNA-seq, an asynchronous parasite culture was grown to 4–7%, at which point only two thirds of the media was replaced with fresh media; one day after this stress, heparin was added to prevent any further invasion and the parasites were sampled at day 3, 5, 7 and 9 following the stress to cover a broad range of gametocyte development without excessive contaminating asexuals. For the maturity assays, heparin was also used and gametocytes were scored daily from day 4 after stress. The maturity of gametocytes was assessed and quantified on Giemsa-stained thin-blood smears. For the exflagellation assays, stage V gametocytes were placed in RPMI with 10% serum and 100 μM xanthurenic acid (Sigma), and the number of exflagellation centres was established on a hemocytometer from 15 to 30 min following activation. Exflagellation rates were calculated by using the red blood cell (RBC) density as well as the gametocytemia (stage V) of each culture and were normalized to WT. To track the viability of the parasites during sex determination, synchronous gametocyte cultures of WT, KO and Δ270–699 treated with heparin as above were followed every 24 h from day 2 to day 6 after stress. Gametocytes were stained with Vybrant Green (1:10,000 dilution in RPMI, Thermo Fisher) and 300 nM MitoTracker Deep Red (Thermo Fisher) in a RPMI culture for 30 min at 37 °C. An unstained control and stained controls with MitoTracker Deep Red and Vybrant Green were prepared with non-parasitized RBCs to establish the negative populations for both fluorochromes. After incubation, samples were washed twice, resuspended in 200 μl of phosphate-buffered saline (PBS) and analysed by flow cytometry on an LSRFortessa flow cytometer (BD Biosciences) with BD FACSDiva software (v.9.0.1). Vybrant Green was excited by a blue laser and detected by a 530/30 filter, whereas MitoTracker Deep Red was excited by a red laser and was detected by a 650/10 filter. 500,000 events were collected for each sample. Percentages of the MitoTracker Deep Red positive gametocytes amongst the Vybrant positive parasites were measured. The data collected were analysed with FlowJo v.10.7 (Tree Star). The gating strategy is shown in Supplementary Fig. 1. For the time course of *md1* expression, synchronous cultures of Md1-2A-GFP treated with heparin as above were followed every 24 h from day 0 to day 8 after stress. Each day, parasites were stained with Hoechst 33342 (1:10,000 dilution in RPMI; Sigma) and 1,000 Hoechst-positive parasites were assessed for GFP expression under a ×40 oil immersion lens on a Zeiss AxioImager Z2. The proportion of bright GFP-positive gametocyte stages was established on 100 Hoechst-positive cells.

### Parasite transfection

*P. falciparum* (NF54) parasites were transfected by electroporating approximately 5% ring-stage-infected RBCs with a total of 50 μg circular plasmid as previously established[27]. After electroporation, transgenic parasites were grown under agitation (50 rpm) and were selected with the appropriate drug. The growth of resistant parasites was monitored by Giemsa-stained thin-blood smears under a light microscope. Mutant lines expressing the transgene episomally (KO complementation, overexpression FL and NT) were continuously selected with 2.5 μM blasticidin (Invitrogen). The mutant lines KO, Δ270–699, Md1-2A-GFP, Md1-3xHA, Md1-TurboID and Δint1 were selected with 2.5 nM of WR99210 (from Jacobus Pharmaceuticals) for 10 days and were subsequently grown without drug selection. Clones were derived from these mutants by limiting dilution in the absence of drug pressure. For each transfection, both mixed population and clones were screened by PCR for the correct integration and the absence of a WT locus.

### Generation of plasmids

The genomic DNA used in all cloning reactions was extracted by using the NucleoSpin Blood kit (Macherey Nagel). Before extraction, erythrocytes of a highly parasitized culture of *P. falciparum* (NF54) were lysed in a solution of 0.1% saponin (Sigma) in PBS, for 5 min on ice, and parasites were harvested by centrifugation (3,600g, 4 °C, 5 min). The PCR reactions were performed by using KAPA HiFi polymerase (Roche) and the colony screening was done with GoTaq DNA Polymerase (Promega) following the manufacturer's instructions. The annealed oligonucleotides used for the cloning of sgRNA sequences were first phosphorylated with T4 PNK for 30 min at 37 °C and were then cloned into the pDC2-Cas9-hDHFR-yFCU plasmid[28] after BbsI digestion by using T4 DNA ligase (overnight at 16 °C). All restriction enzymes, T4 DNA ligase and T4 PNK were purchased from NEB. Fifteen different plasmids were prepared to study the function of *md1* (PF3D7_1438800):

**KO plasmid.** To completely excise the gene via a double crossover homologous recombination event, two donor sequences were designed to target the region upstream of the start codon (homology region 1; HR1) and downstream of the stop codon (homology region 2; HR2). Homology regions were amplified from genomic DNA with primer pairs 202/203 (522 bp) and 116/117 (554 bp), respectively, and sequentially inserted into the plasmid pLN-HAx3 by In-Fusion HD cloning (Clonetech) following digestion with AvrII and EcoRV (HR1) or AflII and BamHI (HR2). A sgRNA targeting the 5′ end of the gene was selected by using the tool available at https://zlab.bio/guide-design-resources[29] and the sgRNA (annealed primers g33/g34) was cloned into the pDC2-Cas9-hDHFR-yFCU plasmid. The resulting parasite line was named KO in which the entirety of the *md1* open reading frame (ORF) was removed, without integration of a resistance cassette.

**Complementation plasmid.** Complementation of the KO was done in trans through the episomal expression of the complete *md1* ORF preceded by 1,538 bp of its putative promoter region. The promoter was amplified from genomic DNA by using primers 209/210 (1,538 bp) and cloned into the pLN-HAx3 digested with ApaI and AvrII (HR1) by In-Fusion HD cloning. This intermediate plasmid was named the pLN-md1prom-HAx3-intermediate. Next the FL ORF was amplified by using as template stage IV/V gametocyte complementary DNA (cDNA) (see below) and primers 211/208 (2,100 bp). This was cloned into the pLN-md1prom-HAx3-intermediate digested with AvrII and NaeI by In-Fusion HD cloning.

**Δ270–699 plasmid.** A different gene-disruption strategy was used to generate the mutant line Δ270–699. In this case, two donor sequences were designed to target the end of the first exon by using primers 287/288 (HR1, 505 bp) and the fourth exon with primers 289/290 (HR2, 474 bp). These products were sequentially cloned into the plasmid pDC2-U6-Cas9-hDHFR[30] with a Gibson assembly strategy following digestion with AatII and EcoRI (HR1) and ApaI (HR2). Insertion of each of the HRs on either side of the drug resistance cassette (hDHFR) present in the plasmid ensured the integration of the latter in the genome of the parasite upon transfection. A sgRNA sequence targeting the beginning of the fourth exon was cloned into the same plasmid by using the annealed primers g92/g93. Given the unexpected phenotype of the Δ270–699 mutant, we assessed the coding potential of the disrupted locus. After integration of the plasmid by double crossover homologous

recombination, the first 808 bp (out of a total of 1,006 bp) of the exon 1 remained undisturbed. In the modified locus, this segment of the *md1* gene was followed by 20 bp of plasmid sequence, a stop codon and the 3′ untranslated regions of the resistance gene cassette encoded in the reverse strand. In the remaining 476 bp at the 3′ end of the gene, the lack of a promoter, and a frameshift that generated six stop codons, eliminated its coding potential. As a result, in Δ270–699 mutants we expect the expression of a truncated form of Md1 with the first 269 aa of the first exon.

**Overexpression plasmids.** For the NT plasmid, to overexpress the N terminus of Md1, we amplified the exon 1 from stage IV/V gametocyte cDNA with primers 211/212 (1,039 bp). This fragment was cloned into the pLN-md1prom-HAx3-intermediate after digestion with AvrII and NaeI by In-Fusion HD cloning. The FL plasmid corresponds to that used for complementation as described above.

**Md1:HAx3 plasmid.** To introduce a triple haemagglutinin (HAx3) tag at the C terminus of Md1 via a double crossover homologous recombination event, two donor sequences were designed to target either the region immediately upstream (homology region 1; HR1) or downstream of the stop codon (homology region 2; HR2). The HR1 was amplified from genomic DNA by using primers 114/115 (630 bp), whereas the HR2 was amplified with primers 116/117 (554 bp). These fragments were sequentially inserted into the plasmid pLN-HAx3 by In-Fusion HD cloning (Clontech) following digestion with NaeI and EcoRV (HR1) or AflII and BamHI (HR2). Three shield synonymous mutations were included in the primer 115 to prevent Cas9 targeting after recombination. A sgRNA targeting the 3′ end of the gene was selected and the sgRNA sequence (annealed primers g31/g32) was cloned into the pDC2-Cas9-hDHFR-yFCU plasmid[28]. Both plasmids, one providing the donor recombination sequence and the other the Cas9, were transfected simultaneously to obtain a selection-marker-free edition of the locus with a scarless insertion of the tag. The resulting parasite line was named Md1-3xHA.

**Md1:T2A:gfp plasmid.** This plasmid was generated from the Md1:HAx3 plasmid, by replacing the 3xHA tag with a T2A:gfp cassette amplified from a template plasmid with primers 147/148. This was assembled with In-Fusion HD cloning following digestion of the Md1:HAx3 plasmid with NaeI and AflII. The same guide plasmid (g31/g32) was used to integrate the T2A:gfp cassette in frame with *md1* at the C terminus. The resulting parasite line was named Md1-2A-GFP.

**Md1:HAx3-TurboID plasmid.** This plasmid was generated from the Md1:HAx3 plasmid, by replacing the 3xHA tag with a TurboID cassette. The latter was amplified from the template plasmid 3xHA-TurboID-NLS_pCDNA3 (Addgene; no. 107171) by using primers 316/317. This was assembled with In-Fusion HD cloning following digestion of the Md1:HAx3 plasmid with AflII and DraIII. The same guide plasmid (g31/g32) was used to integrate the TurboID cassette in frame with *md1* at the C terminus. The resulting parasite line was named Md1-TurboID.

**Δint1 plasmid.** To remove the first intron of *md1*, we designed a 671 bp gene fragment g-block (IDT) that included the last 313 bp of the first exon followed by the beginning of the second exon (317 bp) excluding the intron. The g-block included at its 5′ and 3′ ends 20 bp of overlapping sequences for insertion into the pDC2-U6-Cas9-hDHFR[30] plasmid through Gibson assembly, following digestion with EcoRI and AatII. In addition, the g-block included five synonymous shield mutations to prevent Cas9 targeting after modification of the locus. The sgRNA sequence produced by the annealed primers g94/g95 was cloned into the pDC2-Cas9-hDHFR-yFCU plasmid carrying the g-block. The resulting parasite line was named Δint1.

**CRISPR interference plasmids.** We identified a set of four sgRNAs in the *md1* promoter (spread over the 600 bp preceding the start codon). Each of the sgRNA sequences produced by the annealed primers g52/g35, g54/g39, g55/g41 and g56/g43 was cloned into the sequence-optimized pL7 provided by J. Bryant[31] at the btgZI site. The four pL7 plasmids (15 µg each; selected continuously with 2.5 nM of WR99210) were transfected with pUF_dCas9-3HA-glmS (30 µg; selected continuously with 1.5 µM of Dsm1) also provided by J. Bryant, to generate the parasite line Md1-dCas9-prom. A control parasite line was prepared, which carried the pUF_dCas9-3xHA-glmS alone (Md1-dCas9-ctrl).

A list of all primers that were used for genotyping, cloning and CRISPR–Cas9 guides is provided in Supplementary Table 1.

## qRT–PCR
Cultured stage V gametocytes were spun down at 600*g* for 3 min. After discarding the supernatant, pellets were resuspended in 20× pellet volume of 0.15% Saponin (Sigma) for 5 min on ice, washed with Dulbecco's phosphate-buffered saline (DPBS) once and the pellet was resuspended in 1 volume of DPBS (50 µl) plus 100 µl of TRIzol (Invitrogen) and stored at −80 °C. RNA was isolated according to the TRIzol manufacturer's instructions. RNA was DNase-treated with Turbo DNA-free kit (Invitrogen). cDNA was generated with the High-Capacity cDNA Reverse Transcription Kit (Applied Biosystems) by using provided random hexamers and with the addition of oligodTs (Invitrogen), and was incubated for 1 h at 37 °C, followed by 10 min at 95 °C. The Power SyBR Green Master Mix (Applied Biosystems) was used for qRT–PCR with primers at a concentration of 400 nM in the final mix with the following cycling programme: a single incubation at 95 °C for 1 min followed by 44 cycles (95 °C for 10 s, 60 °C for 60 s). For the lncRNA qRT–PCR the cycling programme was modified to a single incubation at 95 °C for 1 min followed by 44 cycles (95 °C for 10 s, 58 °C for 10 s, 68 °C for 30 s). Fold changes were calculated by using the ΔΔCt method[32]. Sex ratios were measured by using *PfMGET*[9] normalized with the housekeeping gene *uce*[33]. Expression measurements were made with intron-spanning primers and were normalized with *uce*[33]. A list of all primers is provided in Supplementary Table 1.

## Immunofluorescence assay
To measure the proportion of female gametocytes, stage V gametocytes were smeared on a glass slide, fixed with cold 100% methanol, blocked with blocking solution (1% fetal calf serum (Thermo Fisher) in PBS) for 1 h. Parasites were permeabilized for 10 min with 0.1% Triton X (Sigma) and quenched with 0.1 M glycine in PBS (Gibco) for 10 min. The slides were incubated with primary antibodies diluted in blocking solution at room temperature for 2 h; mouse monoclonal anti-α-tubulin (Sigma) and rabbit polyclonal anti-Pfg377 (provided by P. Alano) were used at 1:500. Slides were washed three times in PBS and were incubated with secondary antibodies (1:1,500, Thermo Fisher) in blocking solution for 1 h. Slides were mounted in ProLong Gold Antifade reagent (Thermo Fisher) and were observed under a ×40 oil immersion lens on a Zeiss Axiolmager Z2.

To assess the localization of the Md1-3xHA fusion protein, stage II to V gametocytes were fixed with 4% paraformaldehyde and were allowed to settle on poly-L-lysine coated slides (Thermo Scientific) overnight at 4 °C. Parasites were washed four times with 1× tris-buffered saline (TBS), permeabilized 5 min with 0.2% Triton X in PBS and blocked with 10% goat serum and 3% BSA in TBS for 1 h. Slides were then incubated with primary antibodies diluted in 1% BSA in TBS at 4 °C overnight; polyclonal rabbit anti-ERD2 (BEI, MRA-72) was used at 1:2,000 and monoclonal rat anti-HA (Roche, 3F10) was used at 1:500. Secondary antibodies Alexa-fluor 488/568 anti-rat or anti-rabbit (Thermo Fisher) were used at 1:1,500 dilution in TBS containing 1% BSA and were incubated for 1 h at room temperature. Slides were mounted in Fluoromount-G

with DAPI (Thermo Fisher) and were imaged under a ×63 oil immersion lens on a Zeiss LSM 880 Airyscan confocal microscope with Zeiss ZEN microscopy software (v.2.3 SP1). Images were processed in Fiji[34], the GDSC plugin was implemented on z-stacks by using the Stack Colocalisation Analyser to obtain the Pearson's r coefficient between different channels.

## Shotgun proteomics

120 ml cultures of WT, Δint1 or Δ270–699 parasites were supplemented with 20 U ml⁻¹ heparin three days after stress. Cultures were harvested on day 9 after stress. The pellet was resuspended in ten times the pellet volume of PBS containing 0.15% saponin, incubated for 5 min on ice, washed three times with cold PBS and centrifuged at 4 °C at 3,200g for 5 min. Parasite pellets were then lysed by resuspension in 150 µl of lysis buffer (50 mM Tris pH 8, 150 mM NaCl, 1% NP40, 4 mM EDTA, 1× protease inhibitor cocktail (Sigma-Aldrich)) and incubated for 4 h at 4 °C on a rotating wheel. To improve homogeneity and fluidity, lysates were passed 10 times on a 27 G syringe, sonicated 3 times for 5 s separated by 25 s cooling on ice (Branson Sonifier) before centrifugation at 15,000g for 15 min at 4 °C. The supernatants were harvested and 50 µl of Laemmli 4× (Bio-Rad) and 10 µl of β-mercaptoethanol were added. Protein digestion was done on S-Trap micro spin columns (Protifi) according to the manufacturer's protocol. In brief, protein extracts (300 µg) were resuspended in SDS buffer (5%), reduced with 20 mM of dithiothreitol (DTT) for 10 min at 95 °C and alkylated with 40 mM of iodoactamide for 30 min in the dark. Samples acidified by phosphoric acid (12% final) were diluted in a binding buffer (100 mM triethylammonium bicarbonate (TEABC) in 90% methanol) and proteins were trapped on the columns. Enzymatic digestion was performed by adding 1 µg of trypsin (Gold, Promega) for 2 h at 47 °C. Peptides were eluted with 50 mM of TEABC and formic acid. Peptide samples were fractionated by reverse phase with steps of acetonitrile in eight fractions (High pH Reverse-Phase peptide fractionation kit, Thermo Fisher Scientific). Samples were analysed by nano-flow HPLC-nano electrospray ionization using a Q-Exactive HF mass spectrometer (Thermo Fisher Scientific) coupled with an Ultimate 3,000 RSLC (Thermo Fisher Scientific). Desalting and pre-concentration of samples were performed online on a Pepmap precolumn (0.3 mm × 10 mm; Thermo Fisher Scientific). Peptides were introduced into the column (Pepmap 100 C18; 0.075 mm × 500 mm; Thermo Fisher Scientific) with buffer A (0.1% formic acid) and eluted with a gradient of 2–40% of buffer B (0.1% formic acid 80% CAN) at a flow rate of 300 nl min⁻¹. Eluted peptides were electrosprayed into a Q-Exactive HF mass spectrometer. Spectra were acquired with the Xcalibur software (v.4.2 Thermo Fisher Scientific). MS/MS analysis were performed in a data-dependent mode. Full scans (375–1,500 m/z) were acquired in the Orbitrap mass analyser with a 60,000 resolution at 200 m/z. For the full scans, $3 \times 10^6$ ions were accumulated within a maximum injection time of 60 ms and detected in the Orbitrap analyser. The 12 most intense ions with charges states ≥2 were sequentially isolated to target value of $1 \times 10^5$ with a maximum injection time of 100 ms and were fragmented by higher energy collision dissociation in the collision cell (normalized energy of 28%) and detected in the Orbitrap analyser at 30,000 resolution. Raw spectra were processed by using the MaxQuant environment (v.2.0.3.0)[35] and Andromeda for database search with label-free quantification (LFQ), match between runs and the iBAQ algorithm enabled[36]. The MS/MS spectra were matched against UniProt Reference proteome of *Plasmodium falciparum* (3D7, Proteome ID UP000001450; 2022_01), streptavidin and Δ270–699 Md1 sequence and 250 frequently observed contaminants as well as reversed sequences of all entries. Default search parameters were applied. Oxidation (Met) and acetylation (N-term) were used as variable modifications and carbamidomethylation (Cys) was used as a fixed modification. The FDR was set to 1% for the peptides and proteins. A representative ID in each group was automatically selected by using an in-house bioinformatics tool (Leading v.3.4). First, proteins with the most numerous identified peptides were isolated in a 'match group' (proteins from the 'Protein IDs' column with the maximum number of 'peptides counts'). For the match groups in which more than one protein ID were present after filtering, the best annotated protein in UniProtKB was defined as the 'leading' protein (UniProtKB-GOA, made on 20220317). The MS proteomics data have been deposited to the ProteomeXchange Consortium via the PRIDE[37] partner repository with the dataset identifier PXD035547 and 10.6019/PXD035547. All processed data are included in Supplementary Table 2.

## Western blotting

Cultures were recovered between day 6 and day 10 after stress. Culture suspensions were centrifuged at 600g for 4 min. The pellet was resuspended in ten times the pellet volume of PBS containing 0.15% saponin, incubated for 5 min on ice then washed three times with cold PBS and centrifuged at 3,200g for 5 min at 4 °C. The parasites were resuspended in lysis buffer (50 mM Tris pH 8, 150 mM NaCl, 1% NP40, 4 mM EDTA, 1× protease inhibitor cocktail (Sigma-Aldrich)) (100 µl of lysis buffer for 10 ml of culture) and incubated 4 h at 4 °C on a rotating wheel. Lysates were clarified by centrifugation at 15,000g for 15 min at 4 °C. Proteins were denatured by boiling for 5 min with 1× NuPAGE LDS sample buffer (Fisher) supplemented with 100 mM DTT and separated on 10% SDS–PAGE gels. Gels were transferred onto a nitrocellulose membrane (Amersham) and stained with Ponceau S (0.1% (w/v) in 1% acetic acid). The blots were blocked with 5% (w/v) milk (Bio-Rad) in PBS–0.1% Tween 20 (PBS-T) for 30 min at room temperature then washed three times in PBS-T. Blots were then incubated with a mouse anti-HA antibody (Fisher) in 5% milk in PBS-T overnight at 4 °C, washed three times with TBS-T for 5 min each then incubated with horseradish peroxidase (HRP)-conjugated goat anti-mouse IgG antibody (Bio-Techne) in 1.5% BSA for 1 h at room temperature. Blots were washed three times with PBS-T for 5 min each before developing with the Immobilon Forte WB HRP substrate (Millipore) and were imaged on an Imager 680 (Amersham) or a Chemidoc MP (Bio-Rad). Raw blot images are shown in Supplementary Fig. 2.

## Proximity labelling, proteomics and TurboID analysis

Three replicates of WT and three replicates of Md1-TurboID were prepared. Each replicate consisted of two 40 ml cultures committed two days apart. Two days after stress, cells were washed in biotin-free RPMI (Cell Culture technologies) and the culture medium was replaced by serum-free, biotin-free RPMI supplemented with 1% Albumax, 10 mg ml⁻¹ hypoxanthine and 20 U ml⁻¹ heparin. The medium was changed every day until recovery. Cultures were pooled at day 7 and day 9 post-stress, respectively. Before harvesting, cultures were treated with a 2 h pulse of 100 µM biotin (Thermo Scientific). Cells were processed with saponin and lysis as described above and were washed five times in cold PBS to eliminate excess biotin. Then 80 µl of streptavidin-coated magnetic beads (Fisher) were washed three times in lysis buffer and were supplemented with proteinase inhibitor cocktail then incubated with clarified lysates overnight at 4 °C on a rotating wheel. The beads were subsequently washed twice with lysis buffer, once with 1 ml of 1 M KCl, once with 1 ml of 0.1 M Na₂CO₃, once with 1 ml of 2 M urea in 10 mM Tris-HCl (pH 8.0) and twice with 1 ml lysis buffer[38]. Low-protein binding tubes were used and samples were transferred to new tubes twice during the bead washing process. Biotinylated proteins were eluted by boiling the beads 5 min in 80 µl of 4× Laemmli (Bio-Rad) supplemented with 100 mM DTT. Beads were removed with the magnets and by centrifugation for 5 min at 20,000g and 4 °C. Then 20 µl of lysate, 20 µl of flow through and 5 µl of eluted samples were run on a SDS–PAGE gel as described above. Blots were blocked in a casein buffer (Bio-Rad) and were incubated with HRP-conjugated streptavidin (Cytiva, 1:5,000) in casein buffer for 45 min to 1 h at room temperature, then washed three times with PBS-T for 5 min each before developing as described above. Raw blot images are shown in Supplementary

Fig. 2. For MS preparation, 45 µl of eluted proteins were separated by SDS–PAGE gel using a short (2 cm) migration to avoid biotin contamination. Single pieces of gel were excised for each sample and protein were in-gel digested using Trypsin (Gold, Promega)[39]. MS processing and analysis was done as above. LFQ was used to identify proteins that were differentially enriched between conditions. Only hits found in all Md1-TurboID replicates were analysed. A two sample *t*-test was applied to the $\log_2$(LFQ) values of the WT and Md1-TurboID. In cases for which the WT had only been detected in one replicate, we applied a one-sample *t*-test by using the WT value as the mean against the triplicate values detected in the Md1-TurboID; *P* values were adjusted for multiple testing by using the FDR method. For the calculation of the Md1-TurboID/WT ratios, we imputed the lowest LFQ value detected in the dataset (16) to the missing values in the WT condition, and then calculated the $\log_2$(Md1-TurboID/WT) by using corrected LFQ values. All processed data and calculations are included in Supplementary Table 2. The list of highlighted hits in the volcano plots is described in Supplementary Table 2. The MS proteomics data have been deposited to the ProteomeXchange Consortium via the PRIDE partner repository[37] with the dataset identifier PXD035553 and 10.6019/PXD035553.

### Single-cell transcriptomics

Two methods were used for single-cell transcriptomics, 10x Genomics droplet sequencing was used to phenotype mutants, because it allows the generation of several thousand cells per sample, providing excellent coverage of the sex determination and differentiation process. Smartseq2 was preferred to characterize the Md1-2A-GFP line because by using this method each transcriptome can be associated with information obtained during the cell sort, such as levels of protein expression, given by the GFP intensity signal. This is not yet possible for intracellular fluorescent proteins with droplet-based single cell RNA-seq.

**10x sample preparation, library generation, sequencing and initial mapping.** Four different gametocyte cultures for the WT, KO and Δ270–699 strains were seeded at 2-day intervals, with heparin (20 U ml⁻¹) added after the initial invasion to remove any remaining asexual forms. Cultures were harvested on days 3, 5, 7 and 9 after stress. Gametocytemias were assessed on Giemsa-stained thin-blood smears, and cell density was established on a haemocytometer. The cells from the four time points were pooled at a 1:1:1:1 ratio for each genotype. These pools were counted again with the same method and were seeded at a concentration of 72,000 RBC µl⁻¹ (NF54), 74,000 RBC µl⁻¹ (Δ270–699) and 85,000 RBC µl⁻¹ (KO). Each pool was processed with the 10x chromium platform as described[10] with a target recovery of 8,000 parasite cells per run and by using v3 chemistry. Sequencing was multiplexed on half a lane of the NovaSeq 6,000 S1 flow cell with paired-end (PE) sequencing (26 and 98 bp) and yielded approximately 150 millions reads per run. The data were demultiplexed and mapped to the *P. falciparum* v3 genome sequence (GenedB, June 2021) by using Cell Ranger v.6.0.1.

**Smartseq2 sample preparation, library generation, sequencing and mapping.** Single-cell transcriptomics were conducted as reported previously[10,20] with slight modifications. In brief, gametocyte cultures were spun at 500*g* for 3 min at 4 °C, washed once in DPBS (Thermo Fisher) and stained with 2.5 µM of MitoTracker Deep Red FM (Thermo Fisher) in DPBS for 15 min on ice. Cell sorting was conducted on an FACsAria II cell sorter (BD Biosciences) with a 100 µm nozzle by using BD FACSDiva software (v.9.0.1). Gametocytes were sorted by gating on single-cell events and on the APC and GFP channel. The gating strategy is shown in Supplementary Fig. 1. A stained uninfected RBC control was used for establishing the gating strategy. Single cells were sorted in 96-well plates containing 4 µl of lysis buffer (0.8% of RNAse-free Triton X (Fisher) in nuclease-free water (Ambion), treated with ultraviolet light for 5 min with a Stratalinker UV Crosslinker 2,400 at 200,000 µJ cm⁻², 2.5 mM dNTPs (Life Technologies), 2.5 µM of oligo(dT)

(5′-AAGCAGTGGTATCAACGCAGAGTACTTTTTTTTTTTTTTTTTTTTTT TTTTTTTTT-3′) and 2U of SuperRNAsin (Life Technologies)). Sorted plates were spun at 1,000*g* for 10 s and were immediately placed on dry ice. Plates were heated at 72 °C for 3 min. A reverse transcription mix containing 1 µM of LNA-oligonucleotide (5′-AGCAGTGGTATCAACGCAGAGTACATrGrG+G-3′; Qiagen), 6 µM MgCl₂, 1 M betaine (VWR), 1× reverse transcription buffer, 50 µM DTT, 0.5 U SuperRNAsin and 0.5 µl of Smartscribe reverse transcriptase (Takara) was added to the plates. The total volume of the reaction was 10 µl. The following cycling conditions were used: a single incubation period at 42 °C for 90 min; followed by 10 cycles (42 °C for 2 min, 50 °C for 2 min); before a final incubation at 70 °C for 15 min. A further PCR mix was added to the plates, containing 1× KAPA Hotstart HiFi Readymix and 2.5 µM of the ISO SMART primer[40] and incubated by using the following programme: a single incubation at 98 °C for 3 min followed by 25 cycles (98 °C for 20 s, 67 °C for 15 s, 72 °C for 6 min) and a final incubation at 72 °C for 5 min. Reactions were purified with 1× Agencourt Ampure beads (Beckman Coulter). Amplified cDNA was eluted with 10 µl of nuclease-free water (Ambion). The quality of a subset of cDNA samples was assessed with a high-sensitivity DNA chip (Agilent) with an Agilent 2100 Bioanalyser. Sequencing libraries were prepared by using the Nextera XT 96 kit (Illumina) according to manufacturer recommendations but by using quarter reactions. Dual indices set A and B were used (Illumina) for 192 different index combinations. Libraries were pooled and cleaned up with Agencourt Ampure beads (Beckman Coulter) used at a 4:5 ratio. The quality of the libraries was assessed with the high-sensitivity DNA chip (Agilent) ran on an Agilent 2100 Bioanalyzer. Md1-2A-GFP transcriptomes were multiplexed in two pools of 192 samples and each sequenced on a lane of HiSeq 2,500 with 50 The FASTQ files were obtained after base calling and demultiplexing with Illumina's software. The Nextera adaptor sequences were trimmed with Trim Gallore (v.0.6.5) by using trim_galore -q 20 -a CTGTCTCTTATACACATCT --paired --stringency 3 --length 50 -e 0.1. HISAT2 (v.2.0.0) (ref. [41]) indexes were produced for the *P. falciparum* v3 genome sequence (GenedB, November 2019) by using default parameters. Trimmed, paired reads were mapped to the *P. falciparum* genome sequence by using hisat2 --max-intronlen 5000 p 12. SAM files were converted to BAM by using samtools-1.2 view −b and sorted with samtools-1.2 sort. The read counts were obtained with HTSeq (v.0.12.4)[42] by using htseq-count -f bam -r pos -s no −t CDS by using a custom GTF as described elsewhere[20]. The raw read counts from all single-cell runs are available online[43]. To analyse the splice junctions, sorted bams from the Smartseq2 dataset were processed with portcullis (v.1.2.2)[44] by using --max_length 2000 --min_cov 3. The resulting bed files were sorted and merged with bedtools (v.2.29.1)[45].

**Single-cell analysis.** The 10x data were initially filtered by using Seurat (v.4.0.4)[46]. Cells with less than 1,000 genes and genes present in less than two cells in each dataset were removed from the analysis. The data were normalized by using NormalizeData with the -LogNormalize method, variable features were identified with FindVariableFeatures and the data were scaled with ScaleData. The KO and Δ270–699 datasets were each merged with the WT for a combined analysis. Doublets were identified and were filtered with scDblFinder (v.1.6.0)[47]. After filtration and doublet removal, we recovered 2,076, 2,535 and 2,951 high-quality cells for the WT, KO and Δ270–699 datasets, respectively. For the Smartseq2, the count data were read as single cell experiments (v.1.8.0)[48] and was normalized in scater (v.1.20.1)[4,9] and cells with fewer than 1,000 genes or 25,000 reads were removed from the analysis. Genes identified in more than two cells by at least five reads were kept in the analysis. Filtration resulted in 322 high-quality transcriptomes.

**Classification and ordering of cells, differential expression analysis.** The principal component analysis was conducted within scater (v.1.20.1)[49], and the sexual identity of cells was formally established by

using scmap (v.1.8.0)[50] to map our datasets to an existing *P. falciparum* dataset that contained both asexual and gametocytes from the Malaria Cell Atlas[10,11]. We built a cell index of the reference data and we used the scmapCell function to assign each query cell (from our datasets) to the index cell with which it had the highest cosine similarity. We could then assign the metadata of that index cell to the query cell. We used an assignment cosine similarity threshold of 0.75. This approach oriented the root (progenitor) and two branches (male and female) in our WT-KO, WT-Δ270–699 and Md1-2A-GFP datasets. To order the cells in a trajectory, we implemented lineage and pseudotime analysis by using slingshot (v.1.4.0)[51]. The cells were first clustered by kmeans, before identifying the lineage structure with getLineages, followed by a simultaneous principal curves analysis with getCurves, which fits branching curves to the identified lineages and provides pseudotime estimates for each lineage. The cells were assigned to two developmental paths in all cases going from the progenitor to either the male or the female tip of the branch. A progenitor cell was defined as one that is present in both lineages, a male cell as one absent from the female developmental path and a female cell absent from the male developmental path. Representation of the composition of genotypes along each lineage was generated with a subset of the data with 50 cells selected at random per kmeans cluster to correct for uneven representation of the clusters in each dataset. Differential expression analysis was conducted by using the MAST package (v.1.3.4)[52] in which a hurdle model is implemented to each gene with zlm. The genes with a log fold change superior to 1.25 and a FDR inferior to 1% were considered to be differentially expressed.

## Direct RNA sequencing

$10^9$ NF54 gametocytes (stage II to V) were harvested by centrifugation (500$g$ for 3 min at 4 °C), then lysed with 0.15% Saponin (Sigma) in DPBS on ice for 2 min and washed once with DPBS. The RNA was then extracted with TRIzol Reagent (Thermo Fisher) according to the manufacturer's protocol. RNA quality was checked with the RNA 6,000 Nano Kit (Agilent) with an Agilent 2100 Bioanalyzer. Total RNA (75 µg) was treated with a Dynabead mRNA Purification kit (Thermo Fisher). 500 ng of poly-A RNA were processed with the Direct RNA Sequencing kit from Oxford Nanopore Technologies. In brief, the library is generated by ligation of a 3′ adaptor on the RNA molecules, followed by reverse transcription with Superscript III (Thermo Fisher), purification of the DNA–RNA hybrids with Agencourt RNA XP beads (Beckman Coulter), ligation of the 1D adaptor, purification with Agencourt RNA XP beads (Beckman Coulter) and quantification of DNA–RNA hybrids by using the DNA HS kit (Qubit). The sample was spiked with a control RNA (5%) and was loaded on a MinION flow cell (Oxford Nanopore). Sequencing was performed as recommended with the MinKNOW software (v.19.10.1). Base calling was performed in Guppy (v.3.4.3) and quality control was performed with PycoQC (v.2.5.0.10). After quality control, 639,440 reads with a mean Phred score of 10.63 and a median reads size of 698 bp were recovered. The data were aligned with minimap2 (v.2.17-r941)[53] to the control RNA sequence and Samtools (v.1.9)[54] stats was used to calculate the run error rate, which amounted to 6.72 %. The reads were aligned to the *P. falciparum* genome v3 (GenedB, November 2019) by using minimap2 (v.2.17-r941)[53] with options -ax splice -k14. 79.8% of the reads mapped to the reference genome. Bedtools (v.2.29.1)[45] was used to generate stranded bed files with the genomeCoverageBed command. The coverage analysis and read visualization were performed with the R package Sushi (v.1.24.1)[55]. The coding potential of the lncRNA was tested with CPAT[19].

## Statistical analysis

For all experiments, the number of biological replicates are provided in the figure legends. Statistical analyses were performed in the R environment (v.4.1.0) by using RStudio (v.1.4.1717). The IFA and qRT–PCR sex ratio measurements were first tested for equal variance by implementing an *F*-test with default parameters to compare the variances of the two samples. The samples were then compared by using a student *t*-test with default parameters (two-sided) and by specifying var.equal = TRUE/FALSE in accordance with results of the *F*-test. The mutual exclusivity of expression was tested with a Fisher's Exact Test for count data. The significance codes in the figures: NS, non-significant; *, $P < 0.05$; **, $P < 0.01$; ***, $P < 0.001$.

## Reporting summary

Further information on research design is available in the Nature Portfolio Reporting Summary linked to this article.

## Data availability

Sequencing data that support the findings of this study have been deposited in the European Nucleotide Archive with the accession code PRJEB48349[56]. Processed count data and metadata of single-cell data as well as source data for all figures is available in an online repository[43]. The MS proteomics data have been uploaded to the ProteomeXchange Consortium via the PRIDE partner repository with the dataset identifiers PXD035547[57] (Shotgun) and PXD035553[58] (Interactome). Processed proteomics data is available in Supplementary Table 2.

## Code availability

No new algorithms were developed for this manuscript. Code used for single-cell analysis is available in an online repository[43].

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

**Acknowledgements** We acknowledge the support of the MRI core cytometry and imaging facility, the MGX sequencing core facility and the Montpellier Proteomics Platform (PPM, BioCampus, Montpellier). This work was supported by a starting grant from the Agence Nationale de la Recherche (no. ANR-19-CE15-0007) awarded to A.M.T. This work was supported by the French National Research Institute for Sustainable Development (IRD). A.R.G. is supported by grant no. ANR-19-CE12-0017. A.M.-M. is a recipient of an H2020-MSCA Individual Fellowship (no. 891456). S.A.A. and M.C.S.L. are supported by funding from Wellcome (grant no. [206194/Z/17/Z]). This work was supported by the KIM IBS of the I-Site MUSE. Parasite culture reagents were partially supported by grant no. DEQ2018033199 awarded by Fondation pour la Recherche Medicale to J. J. Lopez-Rubio and by the European Commission (Horizon 2020 Infrastructures #731060 Infravec2 project). We thank Q. Delorme for help with data analysis. We are grateful to J. Bryant for providing dCas9 plasmids and to P. Alano for providing the Pfg377 antibody. We thank F. Ariey, A. Shenoy and M. Lebrun for their input on the manuscript.

**Author contributions** A.M.T. conceived the study. A.R.G. and S.H.A. designed the plasmids and the genetic modification and/or interference strategies and generated the modified parasite lines. A.M.-M., C.B., C.C. and A.M.T. performed cellular and molecular phenotyping experiments. A.R.G., C.B. and C.C. performed the protein work. A.M.T. performed the bulk and single-cell transcriptomic experiments and bioinformatic analyses. A.R.G., M.C.S.L. and A.M.T. secured the funding for the work. A.M.T. wrote the paper with contributions from all authors.

**Competing interests** The authors declare no competing interests

**Additional information**
**Correspondence and requests for materials** should be addressed to A. M. Talman.

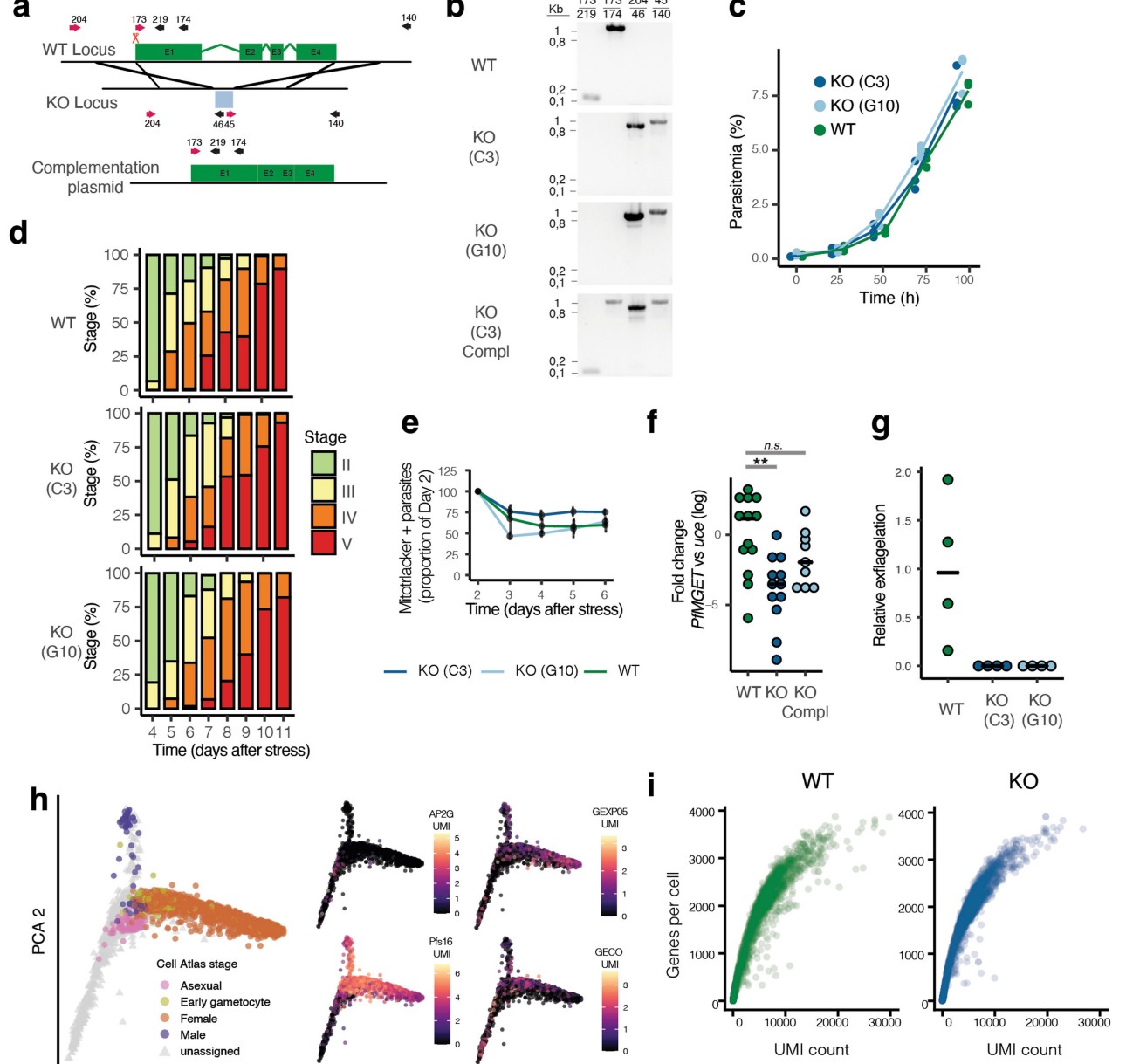

**Extended Data Fig. 1 | Md1 KO genotyping and phenotyping. a**, Gene targeting strategy for complete removal of the *md1* ORF (PF3D7_1438800) via double crossover homologous recombination; the sgRNA site for Cas9 cleavage is symbolized by scissors. **b**, Genotyping PCRs showing the absence of the WT locus (173/219 and 173/174) in either KO clones (C3, G10) and the replacement of the ORF with a generic plasmid fragment (204/46 and 45/140). The complementation strategy included the expression *in trans* from an episome from which a spliced full-length form of *md1* was expressed under the control of the native promoter. The expected sizes for each of the lanes are 137 bp, 1,006 bp, 895 bp and 998 bp, respectively. **c**, Asexual parasite growth of two KO clones is comparable to that of WT parasites (3 biological replicates). **d**, Progression of the WT and KO during gametocytogenesis as monitored by Giemsa-stained thin blood smears also displayed no alteration in the maturation profile of the KO (2 biological replicates). **e**, Time course showing the proportion of gametocytes which retain a mitochondrial potential (stained with MitoTracker Deep Red) from day 2–6 of gametocytogenesis as measured by flow cytometry; asexual parasites were absent from the culture from day 3. Data are presented as mean values +/− SD (6 biological replicates). **f**, Relative qRT-PCR quantification of the male transcript *PfMGET* normalized to the housekeeping gene *uce* in mature gametocyte cultures of WT and KO (C3) and a complemented KO (C3) (biological replicates: n = 13 for WT, n = 12 for KO, n = 9 for KO Compl) (two-sided *t*-test: WT vs KO, *P* = 0.001369; two-sided *t*-test: WT vs KO Compl, *P* = 0.1383).**g**, An exflagellation assay of mature gametocytes shows a complete absence of male gametogenesis in the KO clones (4 biological replicates). **h**, Combined principal component analysis of KO and WT of single cell transcriptomes covering the bifurcation of progenitor cells into male and female branches as identified by maximum cosine similarity onto stages of the Malaria Cell Atlas[10,11] using scmap[50]. Markers of early gametocytes (AP2-G, Pfs16, GEXP05, GECO) are further highlighted. **i**, Quality metrics of all cells in the datasets, showing no specific poor-quality cells in the KO compared to the WT.

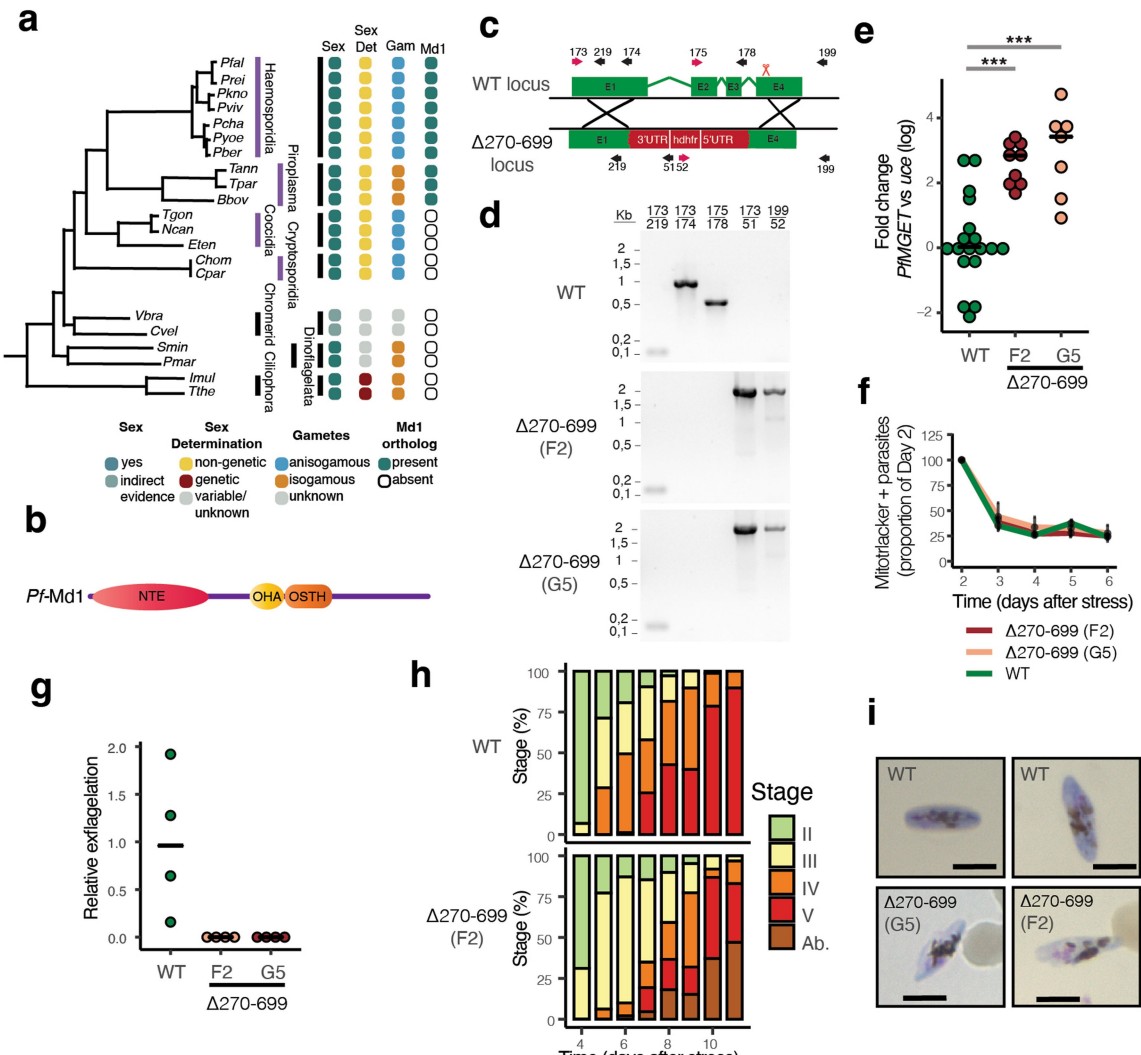

**Extended Data Fig. 2 | Δ270–699 genotyping and phenotyping.**
 **a**, Phylogenetic representation of the Apicomplexa phylum (violet) and alveolate relatives. The phylogeny was adapted from[59]. Sexual strategy in each organism was collected from[1]. Presence of an *md1* ortholog was assessed using OrthoMCL[12] (*Plasmodium falciparum (Pfal), P. reichenowi (Prei), P. knowlesi (Pkno), P. vivax (Pviv), P. chabaudi (Pcha), P. yoelii (Pyoe), P. berghei (Pber), Theileria annulata (Tann), T. parva (Tpar), Babesia bovis (Bbov), Toxoplasma gondii (Tgon), Neospora caninum (Ncan), Eimeria tenella (Eten), Cryptosporidium hominis (Chom), C. parvum (Cpar), Vitrella brassicaformis (Vbra), Chromera velia (Cvel), Symbiodinium minutum (Smin), Perkinsus marinus (Pmar), Ichthyophthirius multifiliis (Imul), Tetrahymena thermophila (Tthe)).* **b**, Schematic of the Md1 protein, including a non-conserved N-terminal extension (NTE) and two conserved domains an OST-HTH associated domain (OHA) and an OST-HTH/LOTUS domain in the C-terminal part of the protein. **c**, Gene targeting strategy for disruption of the *md1* ORF (PF3D7_1438800) via double crossover homologous recombination with integration of the *hDHFR* drug resistance cassette; the sgRNA site for Cas9 cleavage is symbolized by scissors. **d**, Genotyping PCRs showing the disruption of the locus in Δ270–699 clones. The presence of the beginning of the first exon is shown by the primers 173/219 (137 bp) in all strains. The loss of the subsequent WT sequence is

confirmed by the reactions 173/174 (1,006 bp) and 175/178 (656 bp) and correct integration of the resistance cassette is given by 173/51 (1,952bp) and 199/52 (1,915 bp). **e**, Relative qRT-PCR quantification of the male transcript *PfMGET* normalized to the housekeeping gene *uce* in mature gametocyte cultures of WT and Δ270–699 (F2) and Δ270–699 (G5) (biological replicates: n = 19 for WT, n = 9 for Δ270–699 (F2), n = 7 for Δ270–699 (G5) (two-sided *t*-test: WT vs Δ270–699 (F2), $P = 6.09 \times 10^{-7}$; two-sided *t*-test: WT vs Δ270–699 (G5), $P = 0.0007973$). **f**, Time course showing the proportion of gametocytes which retain a mitochondrial potential (stained with MitoTracker Deep Red) from day 2–6 of gametocytogenesis as measured by flow cytometry; asexual parasites were absent from the culture from day 3. Data are presented as mean values +/− SD (6 biological replicates). **g**, An exflagellation assay of mature gametocyte cultures shows a complete absence of male gametogenesis in the Δ270–699 clones (4 biological replicates). **h**, Progression of the WT and Δ270–699 during gametocytogenesis as monitored by Giemsa-stained thin-blood smears displayed a progressive accumulation of stages with an aberrant morphology (Ab.) (2 biological replicates). **i**, Example images of aberrant stages observed in Δ270–699, scale bar 5 μm. This representative observation was made in 3 different independent experiments.

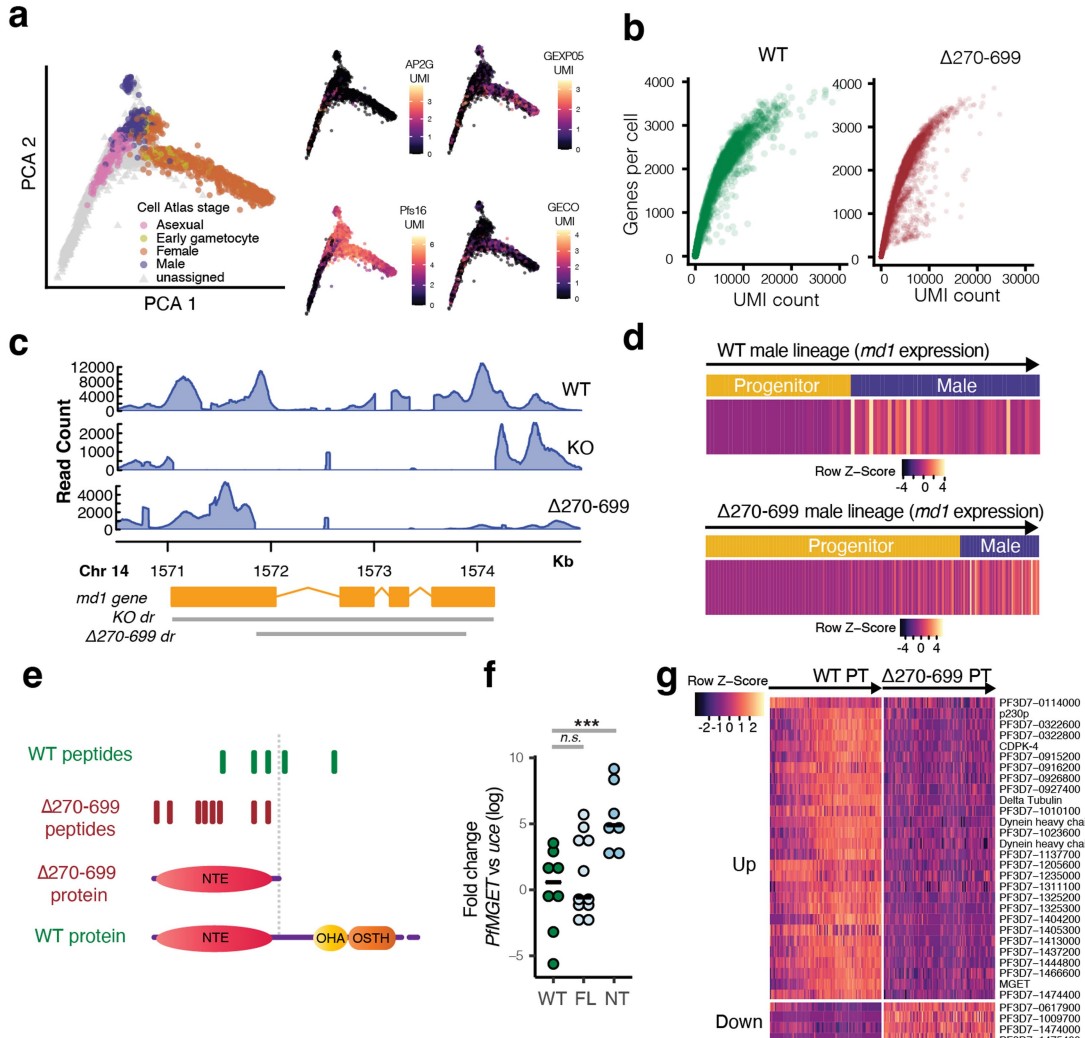

**Extended Data Fig. 3 | Md1 is sufficient to determine a male fate. a**, Combined principal component analysis of Δ270–699 (F2) and WT single cell transcriptomes covering the bifurcation of progenitor cells into male and female branches, maximum cosine similarity mapping with scmap[50] onto stages of the Malaria Cell Atlas[10,11] are displayed for each cell. Markers of early gametocytes (AP2-G, Pfs16, GEXP05, GECO) are further highlighted. **b**, Quality metrics of all cells in the datasets, showing a few extra poor-quality cells in the Δ270–699 compared to the WT probably from aberrant developmentally arrested parasites. **c**, Coverage plots at the *md1* locus of single cell transcriptomes of WT, KO and Δ270–699. Disrupted regions (dr) of the mutants are symbolized by grey bars. **d**, Expression of *md1* in cells during male development in WT and Δ270–699 (F2). **e**, A shotgun proteomics dataset identified peptides in both WT and Δ270–699 demonstrating protein expression in Δ270–699. No peptides were situated in the disrupted region of the mutant (limit indicated by the dashed line). **f**, Relative qRT-PCR quantification of the male transcript *PfMGET* normalized to the housekeeping gene *uce* in mature gametocyte cultures of WT and parasites bearing an overexpression episome of the full-length (FL) or NTE (NT) regions of *md1* (biological replicates: n = 8 for WT, n = 11 for FL, n = 7 for NT) (two-sided *t*-test: WT vs FL, P = 0.4908; two-sided *t*-test: WT vs NT, P = 0.002486). **g**, Genes differentially expressed in the Δ270–699 male branch are displayed in both WT and Δ270–699 lineage with cells ordered in pseudotime. Upregulated and downregulated genes are shown.

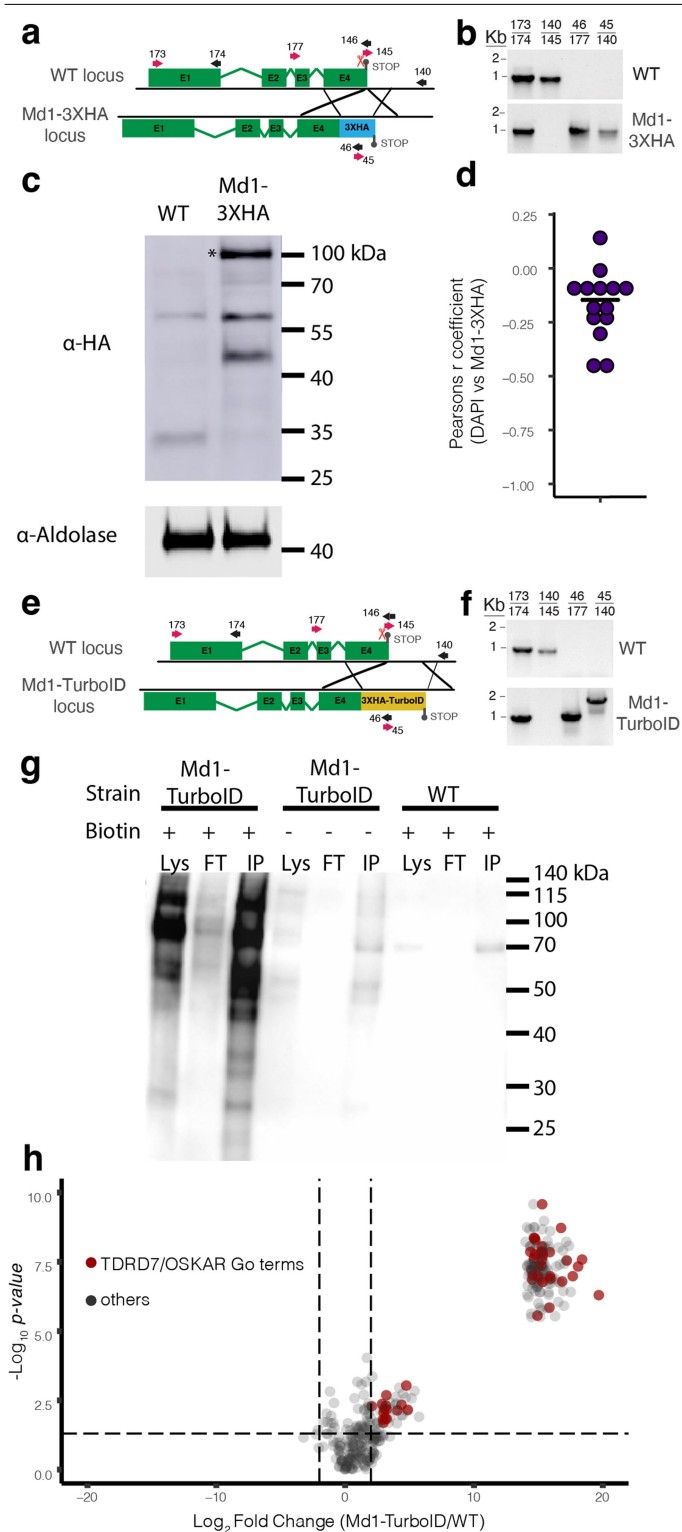

**Extended Data Fig. 4 | Md1 epitope tagging and interactome. a**, Gene targeting strategy for generation of the Md1-3xHA mutant; the sgRNA site for Cas9 cleavage is symbolized by scissors. **b**, Genotyping PCRs showing the correct integration of the construct into the genome. The expected sizes for each of the lanes are 1,006 bp (positive control), 942 bp (WT only); 1,069 bp (mutant only) and 998 bp (mutant only), respectively. **c**, Western blot of WT and Md1-3xHA gametocyte fractions. A band is observed around 100 kDa in the tagged mutant only; the predicted size of the fusion protein is 87 kDa. This representative observation was made in 3 different independent experiments. **d**, 14 Z-stack images of gametocytes (stages III and IV) stained with DAPI and anti-HA antibodies were used to quantify the r coefficient for correlation of the nuclear and Md1 signals, showing that the two signals do not correlate and that Md1 is not present in the nucleus. **e**, Gene targeting strategy for generation of the Md1-TurboID mutant; the sgRNA site for Cas9 cleavage is symbolized by scissors. **f**, Genotyping PCRs showing the correct integration of the construct into the genome. The expected sizes for each of the lanes are 1,006 bp (positive control), 942 bp (WT only); 1,070 bp (mutant only) and 1,958 bp (mutant only), respectively. **g**, Gametocytes of Md1-TurboID and WT were exposed (+) or not (−) to a 2-h pulse of biotin, after which biotinylated proteins were revealed by western blotting with HRP-streptavidin; the input lysate (Lys), flow through (FT) and immunoprecipitated biotinylated fraction (IP) are shown for each condition. **h**, Volcano plot of proteins identified by proximity labelling followed by mass spectrometry of Md1-TurboID vs WT. Following a GO-term analysis of the significant hits, we highlight the proteins included in significantly enriched GO terms that characterize known LOTUS domain containing proteins such as Tdrd7 and Oskar: GO:0072588, "box H/ACA RNP complex"; GO:0000932, "p-body"; GO:0036464, "cytoplasmic ribonucleoprotein granule"; GO:0045495, "pole plasm"; GO:0043186, "P granule"; GO:1990904, "ribonucleoprotein complex". The gene IDs have been highlighted in the Enrichment_Cellularcomponent tab of Supplementary Table 2.

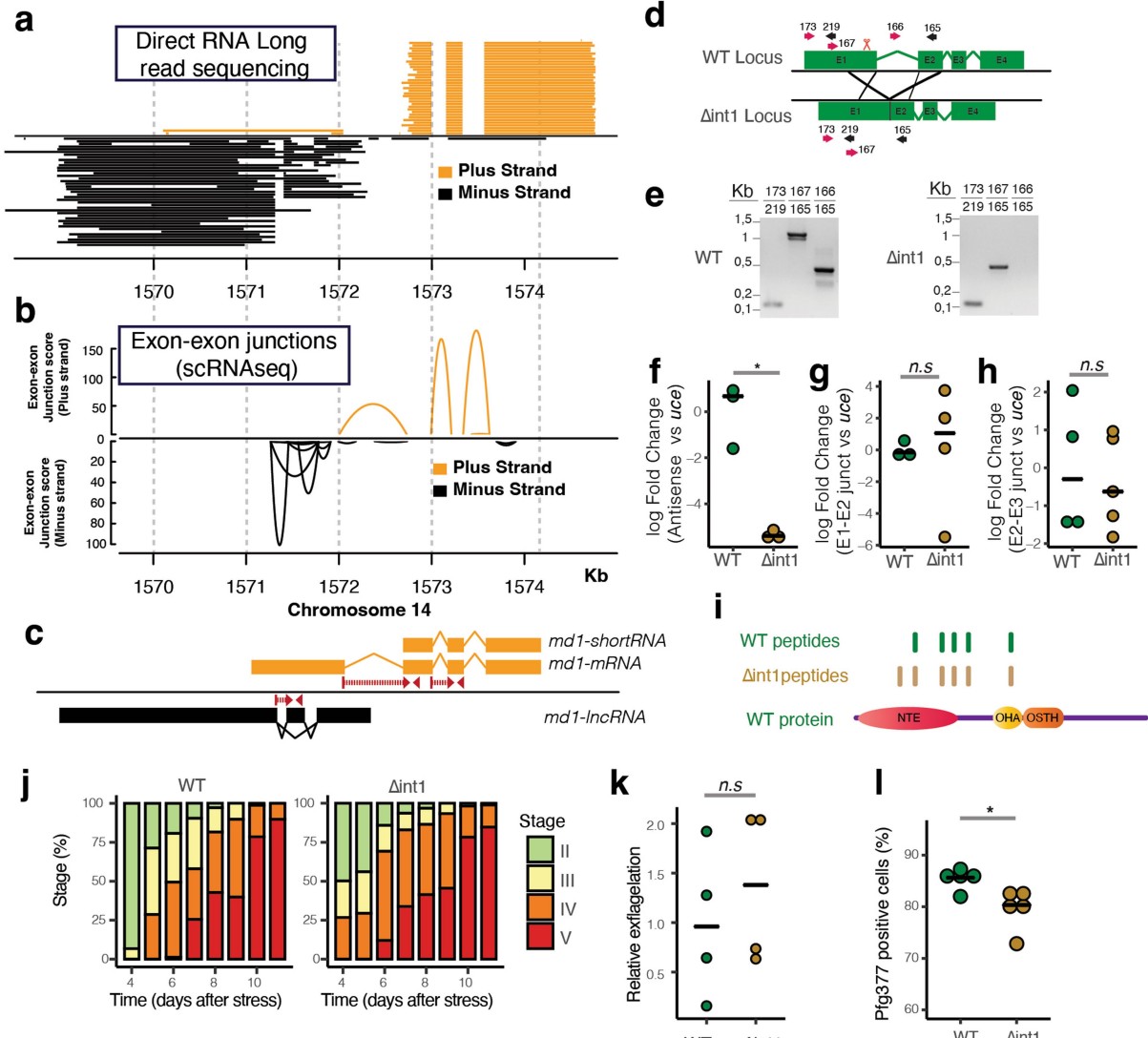

**Extended Data Fig. 5 | Three species of RNA are expressed at the *md1* locus.**
**a**, Reads identified at the *md1* locus in a bulk gametocyte preparation
generated with Oxford Nanopore Direct RNAseq are displayed on each strand.
**b**, Exon-exon junction count from pooled single cell RNAseq gametocyte data.
**c**, Species of RNA transcribed from the *md1* locus. Strand specific primer
positions are indicated in red, intron-spanning portions are dashed. **d**, Gene
targeting strategy for generation of the Δint1 mutant (removal of first intron);
the sgRNA site for Cas9 cleavage is symbolized by scissors. **e**, Genotyping PCRs
showing the removal of the intron in the mutant line. The expected sizes for
each of the lanes are 137 bp (positive control), 1,106 bp WT / 477 bp mutant
(reaction over the locus), and 464 bp (intron specific PCR). **f**, Expression of
*md1-lncRNA* was assessed by qRT-PCR with one primer spanning an intron
within this *lncRNA* to ensure it is strand specific (c). Expression of *md1-lncRNA*
is greatly diminished in Δint1 (3 biological replicates) (two-sided *t*-test:

$P = 0.02043$). **g**, Expression of *md1* was assessed by qRT-PCR with one primer
spanning exon 1 and exon 2 (c) to ensure it is a strand specific measurement
of the full-length transcript (3 and 4 biological replicates) (two-sided *t*-test:
$P = 0.7155$). **h**, Expression of *md1* was assessed by qRT-PCR with one primer
spanning exon 2 and exon 3 (c) to ensure it is a strand specific measurement of
sense transcription (4 and 5 biological replicates) (two-sided *t*-test: $P = 0.5875$).
**i**, A shotgun proteomics dataset identified peptides in both WT and Δint1
demonstrating protein presence in Δint1. **j**, Δint1 displayed no alteration in the
gametocyte maturation profile compared to WT (2 biological replicates). 
**k**, An exflagellation assay of mature gametocyte cultures shows normal levels
of gametogenesis in Δint1 (4 biological replicates) (two-sided *t*-test: $P = 0.5336$).
**l**, Proportion of Pfg377 positive gametocytes (female-specific) by IFA in WT and
Δint1 (5 biological replicates) (two-sided *t*-test: $P = 0.02082$).

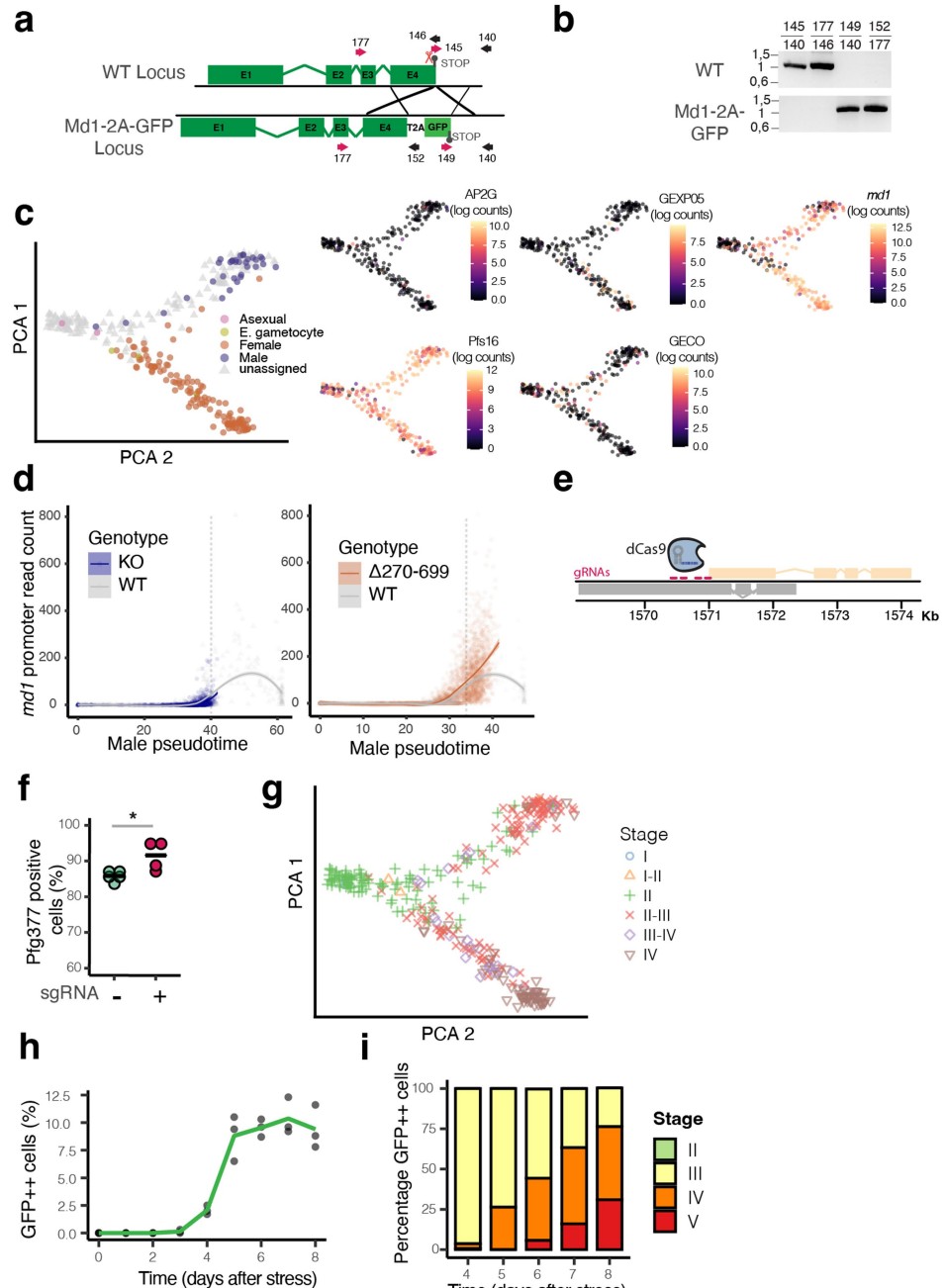

**Extended Data Fig. 6 | Expression at the *md1* locus during sex fate determination. a**, Gene targeting strategy for integration of 2A-GFP at the C-terminus of Md1, via double crossover homologous recombination, to generate the mutant Md1-2A-GFP; the sgRNA site for Cas9 cleavage is symbolized by scissors. **b**, Genotyping PCRs showing absence of the WT locus and the correct integration of the T2A-GFP reporter in the Md1-2A-GFP line. The expected sizes for each of the lanes are 942 bp, 1,023 bp, 1,061 bp, 1,075 bp, respectively. **c**, Combined principal component analysis of Md1-2A-GFP single cell transcriptomes covering the bifurcation of progenitor cells into male and female branches. Maximum cosine similarity mapping with scmap[50] onto stages of the Malaria Cell Atlas[10,11] are displayed for each cell. Markers of early gametocytes (AP2-G, Pfs16, GEXP05, GECO) and unstranded *md1* are further highlighted. **d**, Coverage of the *md1* promoter during sex determination in WT, KO and Δ270–699; the dashed line indicates the separation between progenitors

and males. **e**, Schematic of the CRISPR interference experiment, showing the position of the sgRNAs on the promoter (strawberry). **f**, Proportion of Pfg377 positive gametocytes (female-specific) by IFA in Md1-dCas9-ctrl (without sgRNAs) and Md1-dCas9-prom (with promoter sgRNAs) (5 and 4 biological replicates) (two-sided *t*-test: p = 0.02201). **g**, Correlation analysis with a bulk gametocytogenesis time course[60], each single cell transcriptome is labelled with the time point in the reference time course to which it has the highest Spearman's rank correlation coefficient. It is important to note that early samples in the reference dataset likely contain asexuals as well as stage I gametocytes[60]. **h**, GFP expression in a gametocytogenesis time course of Md1-2A-GFP as measured by the proportion of GFP expressing cells over all parasites (stained with Hoechst 33342) (3 biological replicates). **i**, GFP positive parasite stages were assessed during gametocytogenesis revealing that GFP is expressed from stage III (3 biological replicates).

# Reporting Summary

## Statistics

For all statistical analyses, confirm that the following items are present in the figure legend, table legend, main text, or Methods section.

| n/a | Confirmed | |
|---|---|---|
| ☐ | ☒ | The exact sample size (*n*) for each experimental group/condition, given as a discrete number and unit of measurement |
| ☐ | ☒ | A statement on whether measurements were taken from distinct samples or whether the same sample was measured repeatedly |
| ☐ | ☒ | The statistical test(s) used AND whether they are one- or two-sided<br>*Only common tests should be described solely by name; describe more complex techniques in the Methods section.* |
| ☒ | ☐ | A description of all covariates tested |
| ☐ | ☒ | A description of any assumptions or corrections, such as tests of normality and adjustment for multiple comparisons |
| ☐ | ☒ | A full description of the statistical parameters including central tendency (e.g. means) or other basic estimates (e.g. regression coefficient) AND variation (e.g. standard deviation) or associated estimates of uncertainty (e.g. confidence intervals) |
| ☐ | ☒ | For null hypothesis testing, the test statistic (e.g. *F*, *t*, *r*) with confidence intervals, effect sizes, degrees of freedom and *P* value noted<br>*Give P values as exact values whenever suitable.* |
| ☒ | ☐ | For Bayesian analysis, information on the choice of priors and Markov chain Monte Carlo settings |
| ☐ | ☒ | For hierarchical and complex designs, identification of the appropriate level for tests and full reporting of outcomes |
| ☒ | ☐ | Estimates of effect sizes (e.g. Cohen's *d*, Pearson's *r*), indicating how they were calculated |

*Our web collection on statistics for biologists contains articles on many of the points above.*

## Software and code

Policy information about availability of computer code

| | |
|---|---|
| Data collection | BD FACSDiva software (v 9.0.1) was used during flow cytometry experiments (for both Fortessa and Aria machines).<br>Imaging acquisition- Zeiss ZEN microscopy software (v2.3 SP1)<br>Proteomics acquisition- Xcalibur software (v4.2 ThermoFisher Scientific) |
| Data analysis | Statistical analyses were performed in the R environment (v4.1.0) using RStudio (v 1.4.1717).<br><br>The following analysis packages were used during the analysis (specific functions are specified in the methods):<br>MinKNOW software (version 19.10.1)<br>Guppy (v3.4.3)<br>PycoQC (v2.5.0.10)<br>minimap2 (version 2.17-r941)<br>Sushi (1.24.1)<br>Cell Ranger v6.0.1<br>Trim Gallore (v0.6.5)<br>HISAT2 (v2.0.0)<br>HTSeq (v0.12.4)<br>portcullis (v1.2.2)<br>bedtools (v2.29.1)<br>Seurat (v4.0.4)<br>scDblFinder (v1.6.0)<br>single cell experiments (v1.8.0)<br>scater (v1.20.1)<br>scmap (version 1.8.0)<br>slingshot (v1.4.0) |

MAST (v1.3.4)

Proteomics data analysis
MaxQuant environment (v2.0.3.0)
Leading (v3.4)

Image data analysis
Fiji (v2.3.0) and the GDSC plugin

Cytometry data analysis
FlowJo (v10.7)

Code availibility statement
No new algorithms were developed for this manuscript. Code used for analysis is available in an online repository : https://doi.org/10.5281/zenodo.7211710

For manuscripts utilizing custom algorithms or software that are central to the research but not yet described in published literature, software must be made available to editors and reviewers. We strongly encourage code deposition in a community repository (e.g. GitHub). See the Nature Portfolio guidelines for submitting code & software for further information.

## Data

Policy information about availability of data

All manuscripts must include a data availability statement. This statement should provide the following information, where applicable:
- Accession codes, unique identifiers, or web links for publicly available datasets
- A description of any restrictions on data availability
- For clinical datasets or third party data, please ensure that the statement adheres to our policy

Sequencing data that support the findings of this study have been deposited in the European Nucleotide Archive with the accession code PRJEB48349. Processed count data and metadata of single cell data as well as source data for all figures is available in an online repository (https://doi.org/10.5281/zenodo.7211710). The mass spectrometry proteomics data have been uploaded to the ProteomeXchange Consortium via the PRIDE partner repository with the dataset identifiers PXD035547 (Shotgun) and PXD035553 (Interactome). Processed proteomics data is available in Supplementary Table 2.

# Field-specific reporting

Please select the one below that is the best fit for your research. If you are not sure, read the appropriate sections before making your selection.

☒ Life sciences          ☐ Behavioural & social sciences          ☐ Ecological, evolutionary & environmental sciences

For a reference copy of the document with all sections, see nature.com/documents/nr-reporting-summary-flat.pdf

# Life sciences study design

All studies must disclose on these points even when the disclosure is negative.

| | |
|---|---|
| Sample size | Sample size calculation was not performed, minimum sample size for each experiment was determined empirically based on the standards in the field. |
| Data exclusions | Single cell data was filtered to remove low quality cells based on gene and cell counts. |
| Replication | All measurements were replicated biologically. The number of biological replicates is stated in each figure legend. Two technical replicates were also performed for qPCR measurements. For knock-out mutants, a mutant strain was complemented, or when that wasn't possible two independent clones of the mutants were generated. Data shown from representative experiments were repeated with similar results in at least 3 independent experiments. All attempts at replication were successful. |
| Randomization | Our study did not require randomization. Covariates are not relevant to this study. |
| Blinding | All counting experiments (relative number of females) were blinded by another person than the experimenter before being counted as to avoided experimental bias. |

# Reporting for specific materials, systems and methods

We require information from authors about some types of materials, experimental systems and methods used in many studies. Here, indicate whether each material, system or method listed is relevant to your study. If you are not sure if a list item applies to your research, read the appropriate section before selecting a response.

## Materials & experimental systems

| n/a | Involved in the study |
|-----|----------------------|
| ☐ | ☒ Antibodies |
| ☐ | ☒ Eukaryotic cell lines |
| ☒ | ☐ Palaeontology and archaeology |
| ☒ | ☐ Animals and other organisms |
| ☒ | ☐ Human research participants |
| ☒ | ☐ Clinical data |
| ☒ | ☐ Dual use research of concern |

## Methods

| n/a | Involved in the study |
|-----|----------------------|
| ☒ | ☐ ChIP-seq |
| ☐ | ☒ Flow cytometry |
| ☒ | ☐ MRI-based neuroimaging |

# Antibodies

**Antibodies used**

1- Monoclonal anti-HA antibody produced in Rat; (Roche, 3F10), 11867423001 Sigma, lot: 42155800 (IFA)
2- Monoclonal Anti-α-Tubulin antibody produced in mouse; T5168 Sigma, clone B-5-1-2, ascites fluid; lot : 0000105483
3- Rabbit polyclonal anti-Pfg377 (provided by Pietro Alano) (IFA)
4- anti-HA mouse clone 2-2.2-14 Fisher #11553060 lot #WG327287 (Western)
5- anti-aldolase rabbit polyclonal Abcam #ab38905 lot #GR3242031-4 (Western)
6- anti-mouse HRP polyclonal BioTechne #NBP2-30347H lot #1061068 (Western)
7- goat anti-rat AF488 ThermoFisher #A-11006 lot #2160405 (IFA)
8- goat anti-rabbit AF568 Fisher #10463022 lot #2379475 (IFA)

**Validation**

1- https://www.sigmaaldrich.com/FR/fr/product/roche/roahaha This antibody has been extensively used in the Plasmodium litterature.
2-https://www.sigmaaldrich.com/deepweb/assets/sigmaaldrich/product/documents/371/375/t5168dat.pdf. The tubulin antibody although it was generated against metazoan tubulin has been extensively used in the literature to label P. falciparum tubulin.
3-The Pfg377 antibody was provided by a colleague, his lab and another independent lab have validated the female specificity of the labelling (Schwank, S., Sutherland, C. J. & Drakeley, C. J. Promiscuous expression of α-tubulin II in maturing male and female Plasmodium falciparum gametocytes. PLoS One 5, e14470 (2010)).
4-https://www.thermofisher.com/order/genome-database/dataSheetPdf?producttype=antibody&productsubtype=antibody_primary&productId=26183&version=245
5-https://www.abcam.com/hrp-plasmodium-aldolase-antibody-ab38905.html
6-https://www.novusbio.com/PDFs2/NBP2-30347H.pdf
7-https://www.thermofisher.com/order/genome-database/dataSheetPdf?producttype=antibody&productsubtype=antibody_secondary&productId=A-11006&version=245
8-https://assets.fishersci.com/TFS-Assets/LSG/manuals/mp02764.pdf?_ga=2.113207111.2128744841.1665989921-878038475.1658332919

# Eukaryotic cell lines

Policy information about cell lines

**Cell line source(s)**

BEI

**Authentication**

All cells lines generated in this study were genotyped after initial generation as well as after cloning. Single cell data confirmed the expected genotype of all cell lines tested by this method.

**Mycoplasma contamination**

Cells lines were not tested for mycoplasma infection

**Commonly misidentified lines**
(See ICLAC register)

There are no commonly misidentified lines in our study

# Flow Cytometry

## Plots

Confirm that:

☒ The axis labels state the marker and fluorochrome used (e.g. CD4-FITC).

☒ The axis scales are clearly visible. Include numbers along axes only for bottom left plot of group (a 'group' is an analysis of identical markers).

☒ All plots are contour plots with outliers or pseudocolor plots.

☒ A numerical value for number of cells or percentage (with statistics) is provided.

## Methodology

| | |
|---|---|
| Sample preparation | Cultured samples were stained with 2.5 μM of MitoTracker Deep Red FM |
| Instrument | BD FACsAria II cell sorter - 100 μm nozzle<br>BD LSRFortessa |
| Software | BD FACSDiva software (v 9.0.1) |
| Cell population abundance | (a)Viability assay, parasites are 1-5% of the population, viable cells are 30-100 % of cells<br>(b) Parasite (APC positive) represent  only a few percent of cells (the rest are uninfected erythrocytes) and with the sorted cells 5-10% were GFP positive. |
| Gating strategy | (a)Parasites were gated size, single cells and  on Vybrant green and the proportion of Mitotracker Deep Red positive event recorded<br>(b)Parasites were gated on  size, single cells and APC, and either GFP negative or GFP positive cells were sorted. The index data from the sort is included in the metadata of the single cell RNAseq supplementary file. |

☒ Tick this box to confirm that a figure exemplifying the gating strategy is provided in the Supplementary Information.

