## [Peer Review File · Nature]

Manuscript Title: A transcriptional switch controls sex determination in *Plasmodium falciparum*

Reviewer Comments & Author Rebuttals

Reviewer Reports on the Initial Version:

Referees' comments:

Referee #1 (Remarks to the Author):

This elegant manuscript by Gomes et al. reports the identification of a factor that plays a key role in sex determination in *Plasmodium falciparum*. Understanding the mechanism of sex determination is one of the major current research questions in the malaria field.

The authors provide strong reverse genetics evidence for a role of Md1 as a factor both necessary and sufficient for male sex determination, using knockout, complementation and episomal overexpression lines. The analysis of the mutants relies mainly on scRNAseq, which provided detailed evidence for the transcriptional alterations in the different mutants. Furthermore, an N-terminal domain needed for male sex determination and C-terminal domains involved in male gametocyte development are defined. The manuscript also reports a long non-coding antisense RNA that is only expressed in females, but not in males, in a mutually exclusive manner with the sense md1 transcript. The data is convincing and nicely presented, and the manuscript is well written.

The manuscript largely relies on a previous large genetic screen that enabled the identification of genes necessary for male or female sex determination in *P. berghei* (ref. 7, preprint by Russell et al, Biorxiv). In that preprint, *P. berghei* md1 was identified as a top candidate for male sex determination. The authors used similar scRNAseq approaches to the ones used here to characterize the role of md1 and other candidates.

The manuscript falls short of describing the mechanism by which Md1 (via its N-terminal domain) determines male sex, as there are no experiments addressed to understanding whether it is a transcriptional regulator, a regulator of RNA stability, or plays a different function. The mechanism regulating the expression of the sense or antisense transcripts associated with the male or female fate is also not addressed, and the function of the antisense or the ssRNA transcripts has not been established. The stage of development at which sex determination occurs has not been identified (it may be straight forward to gain this important information, see below). Addressing at least some of these questions would clearly enhance the manuscript, making it even more interesting.

Specific comments.

-The use of "genetic" in the title and the abstract is confusing. The authors classify the mechanism of md1 regulation as a "genetic switch" but two lines later describe sex determination in malaria as "non-genetic". I suggest to use the word "genetic" consistently to refer to information encoded in the primary sequence of the genome, and to refer to the switch in md1 as a "transcriptional switch"

in the abstract, the title and elsewhere.

-The developmental stage at which sex determination occurs is not known. It may occur before sexual commitment, simultaneously or at different post-commitment stages, but based on the results of plaque assays it was proposed that it occurs before the committed schizont stage (Silvestrini et al., 2000; Smith et al., 2000). This should be discussed. The data presented here suggests that sex determination occurs after commitment, but the precise stage is not defined. To gain insight into the precise stage of sex determination, the authors could perform scRNAseq analysis of cultures in which commitment occurs simultaneously within a short time window (e.g., inducing commitment by LysoPC/choline depletion or by using transgenic gametocyte inducible lines). Alternatively, it may be possible to obtain this information without performing additional experiments, simply using stage specific markers (e.g., mapping known markers of schizonts/committed schizonts, sexual rings and stage I gametocytes in the scRNAseq plots to define the bifurcation point). An additional approach to map the stage of sex determination would be to identify the point of md1 activation in publicly available datasets of synchronized gametocyte development transcriptomic time courses.

-The reason why the pKO line produces only females but the delta-Int1 line has a very different phenotype is not resolved, suggesting additional complexity beyond the model proposed for the regulation of md1. While the experiments presented clearly demonstrate separate roles for the N-terminal and C-terminal domains of the protein, neither the absence of the domains involved in male development nor the absence of the antisense transcript explain the absence of females observed in the pKO line.

-The data presented may suggest that gametocytes develop as females by default, and only when md1 is activated they develop as males. Is this the case, or female development requires the activation of alternative genes? Can this be established with the data available? This should be at least discussed.

-Sex determination and sexual conversion are separate processes. To avoid ambiguities, in the second line after the abstract, "sexual conversion" would be more appropriate than "sex". In the first part of the abstract, "sex determination is epigenetic" is not strictly correct; sexual conversion has been demonstrated to be regulated by epigenetic mechanisms, but whether or not sex determination is also epigenetic is not known.

-Md1 has not been described in any published article or annotated in PlasmoDB with this name. Therefore, the ID for this gene should be provided at the beginning of the article, not only in the Methods.

-In the first section of the results, the parasite stages used for the scRNAseq analysis should be described without having to go the Methods. This is a fundamental information to interpret the scRNAseq figures. This is briefly described ("covering gametocyte maturation") only in the next section describing the pKO line.

-To facilitate reading the manuscript, the two methods used for scRNAseq should be consistently

named in the main text and Methods headings. A brief explanation of why different scRNAseq methods are used for different experiments should be provided.

-Fig. 1e (and similar panels in other figures). How were “progenitor” cells assigned to the female lineage or male lineage paths?

-Are the genes in Fig. 1f potential Md1 targets? Do they share some common characteristics? This should be discussed.

-If I understand it correctly, the data for Male KO is from only 4 cells, whereas other columns probably include data from hundreds of cells. This should be explained in the figure legend.

-The schematic of Md1 domains (fig. 2e) should be presented earlier in the manuscript (after the second sentence in page 4). Please indicate the different cell types in fig. 2b.

-Fig. 3a and Ext. fig. 3b. Was a single read covering exon 1 sense identified? Is there an explanation for this? Even if the females are more abundant than males, the ratio of female-specific to male-specific transcripts appears to be unexpectedly high.

-Fig. 3d. It is unclear how the primer spanning an intron was designed, given that multiple alternative splicing events appear to occur for the antisense. The position of the primers should be shown in the figure. Why were primers not designed against the region of the antisense transcript that does not overlap with the sense transcript?

-Page 7. In the sentence “whereas following sex determination, the md1 locus can have two mutually exclusive bistable transcriptional states”, following or preceding? If md1 is the regulator of sex determination, then its activation precedes sex determination.

-Fig. 4d is not cited in the text.

-The parasite line used (NF54) should be indicated in the first section of the Methods.

-Page 13 bottom, the statement does not appear to refer to fig. 1f and 2e.

-Ext. fig. 2b. Is it possible that the promoter driving expression of hdhfr has bidirectional activity, as reported for many malarial promoters, and this affects the regulation of the locus?

Referee #2 (Remarks to the Author):

The mechanism for mating type determination in malaria parasites has been a topic of active investigation for over 40 years, and is extremely likely to be shared across malaria species, including

P. berghei and *P. falciparum*.

An earlier study, with Dr. Talman as a lead author (ref 7), used a forward genetic knockout screen to identify 8 genes that are expressed during early gametocytogenesis of *P. berghei*, 4 of which are essential for male development specifically.

Of these, Md1 was identified as the the most promising candidate for sex determination, based on its early expression following AP2-G induction.

Dr. Talman's team now examines the role of Md1 in sex determination in *P. falciparum* in this manuscript,

Deletion of Md1 had no effect on the asexual growth rate but following sexual commitment only female gametocytes developed, confirming the phenotype previously described in *P. berghei*. Complementation with full length md1 restores the sex ratio to near wt.

Surprisingly, the generation of a md1 truncation mutant that left the promoter and exon 1 intact resulted in a complete loss of female GCs and the remaining GCs only bifurcating into the male lineage.

However, maturation of these GC expressing male markers appears arrested based on both progression along the lineage and the inability to exflagellate.

Indeed, overexpressing the md1 N-terminus in WT cells, greatly increased the fraction of male gametocytes, while overexpression of full length md1 only skewed the ratio slightly.

Analysis of existing bulk RNAseq data and new direct RNA sequencing, identified 3 distinct transcripts from the md1 locus, a sense transcript initiated upstream of exon 1 and both sense and anti-sense transcripts from a promoter inside the first intron.

While deletion of intron 1 completely abrogated anti-sense transcription resulted in a minor shift to more males (15% -> 20% males)

Fusion of a 2A-GFP reporter to Md1 indicates that only the full length transcript is translated but the shorter transcript in females is not.

This study is elegant and carefully carried out.

Barring major issue #1 (see below), it marks a tremendous step forward in our understanding of sex determination in malaria parasites. In particular the observation that overexpression of the Md1 N-terminus can greatly increase males, shows that while the decision was likely made earlier, the mating type is not determined until one or the other isoform of Md1 is expressed.

MAJOR ISSUES:

1. Death of either male or female GCs in the KO or pKO lines would also result in a shift towards 100% of the other mating type.

The authors need to show that the total number of gametocytes formed in the pKO/KO lines remains the same in order to conclusively show that md1 determines the mating type rather than mediates survival of either males or females.

In other words, mutations in Md1 need to turn what would have been females into males (pKO) or vice versa (KO) rather than causing them to die or arrest.

2. Using plaque assays Silvestrini and colleagues (PMID: 11128797) demonstrated quite convincingly that the mating type decision is already established in sexually committed schizonts formed at the end of the previous cycle.

Failure to reference this landmark paper deprives the reader of a crucial piece of context and needs to be thoroughly addressed in the context of the role of Md1.

Since Md1 isn't expressed until around day 3 of gametocytogenesis, (after AP2-G, which is expressed in committed schizonts and during the first 2 days of gametocytogenesis (PMID: 30478286).

So while mating type isn't locked in until expression of the different isoforms of Md1 occurs, which isoform is going to be expressed is determined previously.

This is analogous to AP2-G, which depends upstream factors for its expression state but once that state is established the asexual/sexual differentiation is locked in.

3. Gametocyte generation methods need to be specified in much greater detail. Since methods are not included in the word count there is no reason to keep them so brief.

4. Calling all cells "progenitors" prior to bifurcation of the male/female lineages is unnecessarily vague, but the inability to distinguish them along the male/female axis isn't the only component that can be used to differentiate progenitors.

Many "Progenitors" are clearly early gametocytes, based on their expression of early gametocyte markers.

And since heparin was used to limit asexual growth, these should also include asexuals during at least the first 48 hours of gametocytogenesis, since heparin interferes with merozoite invasion.

At minimum, progenitors should be labeled as early gametocytes or trophozoites/schizonts to clear this up.

The stage composition of the progenitors should be specifically indicated and it needs to be clear to the reader that many progenitors are already gametocytes.

5. At what day of gametocytogenesis does md1 start being expressed? This can readily be determined using synchronous induction of gametocytes using the 2A-GFP line.

6. The authors should comment on known functions of LOTUS/OST-HTH domains and how these might be involved driving the transcriptional programs underlying each mating type.

7. The fact that nearly all GCs are male when Md1 is truncated after exon 1 but only 20% are male when full length md1 is expressed suggest that the region of Md1 protein encoded by exons 2-5 negatively regulates the region of Md1 encoded

8. The authors show md1 is essential for sex-determination and that a bistable switch exists that leads to two distinct transcriptional states at the md1 locus.

However, that doesn't mean that md1 is the bistable switch or part of the bistable switch.

Indeed, the fact that deletion of intron 1 had only minimal effects on sex ration indicates that md1 is downstream of the switch.

That said, the fact that Md1 truncation or complete deletion switches the sex ratio one way or another shows that this decision manifests and locks in only upon expression of md1.

MINOR ISSUES:

Use of pKO for partial KO is confusing since it looks like a plasmid name. a more descriptive name like (like md1 Δ 270-699) should be used.

Fig1C which DHC? There are several with sex specific expression.

Fig S1e,S2d,S2k: dont normalize to female marker (over-estimates the change since they aren't independent but anti-correlated, use uce instead)

Fig S2i: indicated the deleted regions for pKO and KO

How does the male trajectory differ in pKO?

RNAs: what are the approximate 5' & 3' UTRs for md1-mRNA & md1-ssRNA ? are md1-ssRNA or md1-lncRNA polyadenylated?

Please chose another name for the exon 2-4 sense transcript. md1-ssRNA looks like single-stranded RNA not short-sense.

Fig. S3a: reads from male vs female Lasonder is from much later in GC development

Fig. S3b: sense reads from exon 1 are much lower.

Fig. 3c show positions of primers used in 3d.

Fig. 3c The relative abundance of the full length vs ssRNA transcripts should also be shown.

(qRT-PCR with primers in exons 2-5 (ssRNA+mRNA) vs exon1 (mRNA) with sense specific primers used for RT)

Δ intron1: "promoter of md1-lncRNA spans intron 1" cant infer length of the promoter but instead refer to speak to TSS

What are the cells shown in Figure 4?

Plot ap2-g expression on cell fate plots

plot pfs16 on male female

4B middle panel doesn't support statement "in the male lineage, the md1 promoter first becomes active before cells can be unambiguously identified as males and is coupled with protein expression."

Just because you can't distinguish their fate transcriptionally doesn't mean that the decision has not already been made but hasn't resulted in transcriptional differences.

Referee #3 (Remarks to the Author):

In malaria parasites, sexual reproduction coincides with transmission to the vector host, and therefore the understanding of the mechanisms of sex determination and differentiation in *Plasmodium falciparum* are object of intense research. Sex in these organisms is determined

epigenetically, i.e., male and females are genetically identical. The events that regulate the differentiation of sexually committed parasites into either male or female pathways are not well understood, and this paper represents a significant advance in this direction. In particular, the manuscript suggests that Md1 is the key gene in determining the gender of *Plasmodium falciparum*, and proposes an association of the male-determining function with the N-terminus of the Md1 protein, whereas the C-terminus may be involved in male developmental regulation. Moreover, it is suggested that sex determination involves a bistable genetic switch at the Md1 locus where an alternative version of Md1 would produce a lncRNA that ensures silencing of the male fate in female cells.

The manuscript is overall well written and well presented, and tackles an important and old question, both in an applied and fundamental perspective. The approaches used are valid, and data is of high quality. The conclusions are sound, but I have a number of concerns and request for clarifications, described below point by point.

1) How does this manuscript link to a Bioarchives preprint that also describes md1 as one of the ten genes involved in sexual determination/differentiation in *P. berghei*? The main author of the manuscript is one of the first authors of this manuscript. The pre-print describes a screen for genes required for the formation of male and female gametocytes in *P. berghei*. The preprint also uses single-cell transcriptomics to map the sexual differentiation events of lineages at high temporal resolution. Md1 is identified as (one of the) male-determining gene and highlights the domain LOTUS/OST-HT. This preprint describes md1 but also identifies a second locus, md2, as potential male-determining, since disruption leads to a complete loss of cells expressing male markers. In particular, md1 is shown to be the most noticeable AP2-G responsive gene to be upregulated early in the sexual pathway (which I guess was the reason why the authors chose to work on md1 in the present manuscript?). Therefore, there is (at least conceptual and technical) some overlap with the current paper, and it would be important to clarify how the two papers articulate.

2) The gene md2 (for which no detailed information is given in the preprint except that it is not conserved outside *Plasmodium*) appears to have a role in sex (male) determination and I wonder how these findings fit in the model proposed by the authors that md1 is necessary and sufficient to induce a male cell fate. For example, key information could be obtained if some of the experiments of the current paper were performed in a md2 background.

4) I have some reservations regarding the way phenotypic sex was determined, because as far as I understood, markers have been used to infer sex of the gametocytes, but not functional sex (crosses with opposite sex). Because sex is not genetic, these markers correspond to 'sex-biased genes' (that have been previously identified). Without a clear determination of the phenotypic sex of the gametocytes, using genetic crosses which reveal the 'real' functional sex, the whole interpretation of the results may be flawed, biased and/or incomplete. This is because the cascade of genes involved in male and female development ('sex-biased genes') may be decoupled with the 'functional' sex (for example, lines may be expressing female markers – i.e., be feminized – but still be functionally males, in the sense they fuse with female gametes). For example, can you exclude that the KO is simply affected in some parts of the sex-differentiation cascade (i.e., the line may be 'feminized' based on the markers), but still be phenotypically male (i.e., would be capable of fusion with female gametes)? Controlled genetic crosses appear to be feasible in *Plasmodium* because I see they were performed in the preprint (Russel et al).

5) Did I understand correctly that two mutants were produced? KO (full) and pKO (partial)? But only

one full KO? Considering the effect of KO and pKO is different in terms of phenotype, wouldn't you need a couple of independent mutant alleles to validate the link between phenotype and genotype? Or do you consider that KO and pKO are 'alleles'?

6) If Md1 is involved in sex determination, one would expect that the protein Md1 should be produced in males gametocytes (but not female)? The manuscript is missing md1 protein analysis., using a proteomic dataset would allow to verify that md1 is only present in male gametocytes and provide additional evidence that this gene is indeed exclusive to the male pathway. Perhaps sex-specific proteomic datasets are already available and could be explored.

7) 'Timing' of male fate and expression of the md1 needs clarification. The authors state that md1 mRNA is present after the fate is determined "exon 1-2 junction (which can only arise from md1-mRNA) was exclusively expressed immediately following bifurcation into the male branch". However, if md1 is involved in sex determination, it would be expected that the transcript (and protein) should be present before the bifurcation already, in the cells that will become male, and not once the fate has been already determined. Indeed, it is stated later in the text "In the male lineage, the md1 promoter first becomes active before cells can be unambiguously identified as males and is coupled with protein expression." however, I don't see where is the figure/results that support this statement, perhaps I missed it?

8) the authors remove intron1 (Δ int1 line) and confirm that the intron is indeed important for md1-lncRNA transcription. I could not find the control showing that md1 is expressed to normal levels (and that the md1 protein is there – see my comment below about proteomics) in the Δ int1 line?

9) I may be confused here, but if Md1 is already differentially expressed in the progenitor cells (upregulated ? see fig 1f), how is it possible that a proportion of the progenitor cells will become female? Does this mean there are genes involved in the female fate that override md1 in these cells? Again, knowledge about the presence of the md1 protein in progenitor vs male and female cells would be perhaps useful. Importantly, the authors cannot exclude that other genes are involved in the pathway. As indicated in the preprint, several other genes are identified as related to sex determination, so I would be careful in not overstating Md1 as necessary and sufficient for sex determination, it is probably one (key) actor in the pathway.

10) The authors state that "md1-lncRNA is not the main driver of transcriptional silencing of the md1 promoter in females." Do you have an idea of what is the target of the md lncRNA in females? The authors have several datasets on genes involved in the sex determination pathways, could any of the genes that are silenced in females (strongly upregulated in males) be the target?

11) I find a bit confusing that a range of different male and female markers are used throughout...is there a way to simplify this? Perhaps listing in the methods clearly which are the male and female markers. Actually, I was wondering why different markers are used for different experiments? I would have thought it is better to have several male and female markers per experiment (instead of only one marker per experiment, and different markers in different experiments). What was the basis of the choice of the different markers in each experiment? Also, I would like to have some clarification concerning the use of PG377 as a female marker, as this marker, if I understand right, only works for late stages of gametogenesis. Could the fact that this marker does not capture female fate in early stages affect your results?

Finally, the question that is not answered and that would be needed to fully apprehend the sex determining pathway is what actually triggers the switch (what triggers the production of the sense

transcript 'md1-mRNA' in males and lncRNA transcript in females). Perhaps one of the genes that are described in the preprint? Although I am not requesting the authors to do this, but I think it is important to emphasize that Md1 is involved in sex determination, but may not be the one and only 'master' sex-determining gene.

Other comments:

Figure 1c) what is the different between the upper and lower panels?

Figure 1f) shows several DE genes there, among them there is md1. What are these other genes?

Are they also involved in sex determination?

Section "Md1 is necessary to determine a male fate". If I understood correctly, there is a reduction of proportion of mature males, however, if this gene is the male master sex determinant, then the effect of the KO should be seen early too. Is this the case?

Extended fig 1e – In order to obtain a mean logFC of about 0.5 in the WT (or at least a positive value) you would need to have more male than female marker (because normalization is performed based on male/female marker?). If this is the case, there is a strong male bias in the WT culture. However, in the text it is stated that there are about 85% female (Pfg377)-positive gametocytes in the wild type (because there is a strong female-bias). Isn't there an inconsistency somewhere, or perhaps I am misunderstanding?

It would have been useful to have northern blots to validate the splice variants of md1 ?

Related to this, I don't understand why the authors don't normalize the QPCR using a normalization gene (i.e. that has a constant expression, regardless of the sex). This needs to be clarified.

Extended fig 1d – for consistency, please present data also for KO G10.

"To avoid the pitfalls of distinguishing gametocyte sex morphologically"

Statistical analysis to support the sentence "The proportion of Pfg377-positive gametocytes was 85% in the WT (fig1b), indicating a typical female-biased sex ratio. In contrast, KO gametocytes were almost exclusively Pfg377-positive (97%), indicating very few, if any, males in the population." is required. In the legend of the figure it is mentioned that "n=4" but no indication of numbers of cells nor the statistical tests that were performed. Is 85% vs 97% significantly different? What does the n=4 mean in the legend? Do I understand correctly that you used 4 different mutant lines (independently mutated/alleles)? How many gametocytes were counted in each line? What do the asterisks correspond to? Detailed stats explanations, including what you call 'n' are also missing in legends for example Extended figure 2k, 2d, 2e, etc.

"Taken together, this suggests that md1-lncRNA is dispensable for male determination and its knockdown may even enhance bifurcation to a male fate." – How do you explain this observation that there is even an increase in male fate in absence of md1-lncRNA?

Perhaps a more philosophical question is why we use the term 'sexes' and not 'mating type' – given that these organisms appear not to be clearly anisogamous/oogamous (eggs and sperm), so no asymmetry between gametes?

Author Rebuttals to Initial Comments:

Dear Dr Tobin Kahrstrom,

I would like to thank you and the reviewers for your thorough assessment of the manuscript.

We have addressed each of the reviewers' comments specifically and also enhanced the manuscript to respond to your general appreciation in the following ways:

That said, the referees are unclear about the correspondence between this paper, and the related paper on bioRxiv (which shares several authors). Given that this paper does indeed impinge on the novelty of the manuscript, we need to ask about its status, and you would need to clarify how this paper represents a sufficient advance given the clear overlap in subject matter.

The rodent malaria manuscript is still in evaluation. We do not think it impinges on the novelty of the current manuscript since this genetic screen identifies genes that are linked to sexual development, and phenotypes are realised at the end point of differentiation and as such cannot differentiate between developmental and cell fate defects. Furthermore, this initial screen was conducted in the rodent malaria model and not on the significantly more challenging but relevant human malaria species *P. falciparum*. It is noteworthy that sexual development is quite divergent between these two species.

The paper submitted to Nature is a major advance in the field because it (i) identifies the molecular basis of sex determination in human malaria parasites (ii) specifically maps the defect in the determination step and provides a roadmap for early stages of development following determination (iii) uncovers the regulatory architecture surrounding a master regulator of sex determination providing a model for non-genetic sex determination in basal eukaryotes (iv) highlights the main player in a pathway essential to transmission in the deadliest human malaria parasite and the biological pathway through which it is likely to function, opening the road to potential translational research. None of these are covered in the paper on bioRxiv.

Related to this, we do feel that further development of mechanism, along the lines suggested by referee #1 in particular, would strengthen the case for further consideration at Nature.

We agree that these suggestions would make our study even more compelling, and have now added several new experiments to address key questions highlighted by reviewer 1. Specifically we have (i) performed epitope tagging and localised the protein to cytoplasmic foci (ii) produced an interactome of the protein using proximity labelling and proteomics to gain further insight into its molecular function; (iii) used a crispr interference approach to further understand the regulation at the *md1* locus; (iv) conducted further experiments to understand the timing of sex determination. These experiments and their incorporation in the manuscript are detailed below.

In addition, it seems important to provide compelling evidence to rule out gametocyte death as an explanation for the shifts observed in mating type.

We have done further experiments and analyses and have excluded the possibility of cell death as the cause of our observed phenotypes. A detailed description is provided below.

Furthermore, referee #3 raises an important concern relating to the reliance on phenotypic markers, and requests further experiments to validate correspondence between phenotype and genotype.

We agree that this is an important point and have described below how our single cell phenotyping approach precludes the possibility that our mutants simply are deficient for a few specific sex markers, since we see canonical sex-specific transcriptional signatures in our mutants. The expression programme of both sexes differs by several hundred genes, underlying the very different developmental cellular programmes of a male and female gametocyte.

Referees' comments:

Referee #1 (Remarks to the Author):

This elegant manuscript by Gomes et al. reports the identification of a factor that plays a key role in sex determination in *Plasmodium falciparum*. Understanding the mechanism of sex determination is one of the major current research questions in the malaria field.

The authors provide strong reverse genetics evidence for a role of Md1 as a factor both necessary and sufficient for male sex determination, using knockout, complementation and episomal overexpression lines. The analysis of the mutants relies mainly on scRNAseq, which provided detailed evidence for the transcriptional alterations in the different mutants. Furthermore, an N-terminal domain needed for male sex determination and C-terminal domains involved in male gametocyte development are defined. The manuscript also reports a long non-coding antisense RNA that is only expressed in females, but not in males, in a mutually exclusive manner with the sense md1 transcript. The data is convincing and nicely presented, and the manuscript is well written.

The manuscript largely relies on a previous large genetic screen that enabled the identification of genes necessary for male or female sex determination in *P. berghei* (ref. 7, preprint by Russell et al, Biorxiv). In that preprint, *P. berghei* md1 was identified as a top candidate for male sex determination. The authors used similar scRNAseq approaches to the ones used here to characterize the role of md1 and other candidates.

The manuscript falls short of describing the mechanism by which Md1 (via its N-terminal domain) determines male sex, as there are no experiments addressed to understanding whether it is a transcriptional regulator, a regulator of RNA stability, or plays a different function. The mechanism regulating the expression of the sense or antisense transcripts associated with the male or female fate is also not addressed, and the function of the antisense or the ssRNA transcripts has not been established. The stage of development at which sex determination occurs has not been identified (it may be straight forward to gain this important information, see below). Addressing at least some of these questions would clearly enhance the manuscript, making it even more interesting.

We thank the reviewer for all these interesting suggestions, many of which are part of this improved manuscript or the subject of future studies in our laboratory.

Specifically, we have:

-Generated an HA-tagged line of Md1 and shown it does not localise to the nucleus but rather to cytoplasmic foci suggesting it is not a transcription factor (Fig. 2g, Extended Data Fig. 4a-d).

-We further generated a line where Md1 is associated with a biotin ligase (Md1-TurboID) in order to conduct proximity labelling, and produced an interactome of Md1 (Fig. 2h, Extended Data Fig. 4e-h). Remarkably we find a very significant enrichment for RNA binding proteins and elements of RNP granules suggesting, in agreement with the localisation, that the function of Md1 may be

exercised through RNA-containing granules in differentiating gametocytes. This finding is particularly notable as it is reminiscent of the already established function of p-bodies, which are present in the female lineage and have a role in parasite host transition during transmission to mosquitoes. Importantly LOTUS-containing proteins are known to associate with proteins present in RNP granules. In the Md1 interactome we have identified several Md1-interacting dead-box helicases; these helicases are thought to mediate indirect LOTUS-protein/RNA interactions in other systems (e.g. *vasa*, *TDRD9*), suggesting a potentially conserved association. This finding highlights the potential cross-kingdom conserved molecular function of this domain and its role in developmental decisions through RNA biology albeit in different cellular processes. Altogether our data provides the first indication of how sex may be determined on a molecular level. These results are described in the text (Page 5, lines 2-17).

- To clarify the regulation of expression at the *md1* locus, we have elaborated on our interpretation of the data (Page 9, line 16-34). We hypothesise that the two mutually exclusive states of the locus are underpinned by mutually exclusive binding of regulatory factors at the locus. Since the phenotype of Δ int1 line is a slight shift to males, we interpret it to mean that the regulation of the locus and especially its female-state is likely dependent on the region spanning intron1+exon-2-4 and not just on intron1. We use CRISPR interference to block access to the promoter of *md1* and find that this line shifts to a more female ratio (Extended data Fig.6e,f), further supporting the notion of a competition between regulatory platforms at the locus. Altogether our new data brings a more robust understanding of how the locus may be regulated. As reviewer 3 also mentions this point, we are very much thinking about which players may be involved in regulating the locus but this will be the next line of inquiry in a future research effort.

-Finally, we have included further experiments to evaluate the timing of sex determination as suggested. Using our Md1-2A-GFP line, we used GFP fluorescence as a proxy for Md1 expression and have timed the initial expression of Md1 during gametocytogenesis (stage III) (Extended data Fig. 6g,h). We also provide an enhanced discussion of the timing of sex determination in *Plasmodium* as suggested by all reviewers (Page 10, line 13- Page 11, line 10).

Specific comments.

-The use of “genetic” in the title and the abstract is confusing. The authors classify the mechanism of *md1* regulation as a “genetic switch” but two lines later describe sex determination in malaria as “non-genetic”. I suggest to use the word “genetic” consistently to refer to information encoded in the primary sequence of the genome, and to refer to the switch in *md1* as a “transcriptional switch” in the abstract, the title and elsewhere.

We agree with the reviewer and have amended our nomenclature to non-genetic determination and transcriptional switch.

-The developmental stage at which sex determination occurs is not known. It may occur before sexual commitment, simultaneously or at different post-commitment stages, but based on the results of plaque assays it was proposed that it occurs before the committed schizont stage (Silvestrini et al., 2000; Smith et al., 2000). This should be discussed. The data presented here suggests that sex determination occurs after commitment, but the precise stage is not defined. To gain insight into the precise stage of sex determination, the authors could perform scRNAseq analysis of cultures in which commitment occurs simultaneously within a short time window (e.g., inducing commitment by LysoPC/choline depletion or by using transgenic gametocyte inducible lines). Alternatively, it may be possible to obtain this information without performing additional experiments, simply using stage specific markers (e.g., mapping known markers of schizonts/committed schizonts, sexual rings and stage I gametocytes in the scRNAseq plots to

define the bifurcation point). An additional approach to map the stage of sex determination would be to identify the point of *md1* activation in publicly available datasets of synchronized gametocyte development transcriptomic time courses.

Using ours as well as other published datasets (Poran et al. 2017, Reid, Talman et al. 2018), and a yet to be published committed ring data set, we have failed to find any signature or heterogeneity in schizonts, committed rings, or young gametocytes suggestive of a transcriptional sex-determined signature in those stages. Our data consistently shows that there is a gametocyte transcriptional state in early gametocytogenesis that is neither male nor female, strongly suggesting that determination occurs after commitment.

As noted above, we have conducted an additional experiment with the *Md1-2A-GFP* line (Extended data Fig. 6g,h) to understand the exact timing of determination in a time course and find that *Md1* is present from stage III onwards. Although this doesn't mean that there isn't a transcriptionally-silent predetermination state as the Silvestrini *et al.* and Smith *et al.* data would suggest. An interesting alternative explanation is that the plaque assay isolated parasites within the same microenvironment leading to environmentally-determined commitment to a particular sex without the cell fate decision having been realised at the schizont stage. We also note that same cycle commitment has recently been demonstrated (PMID: 30478286) and that the point of determination in this pathway is not known. We have included a more elaborate discussion and contextualisation of the timing of the cell fate decision in the discussion (Page 10, line 14-Page 11, line 10).

-The reason why the pKO line produces only females but the delta-*Int1* line has a very different phenotype is not resolved, suggesting additional complexity beyond the model proposed for the regulation of *md1*. While the experiments presented clearly demonstrate separate roles for the N-terminal and C-terminal domains of the protein, neither the absence of the domains involved in male development nor the absence of the antisense transcript explain the absence of females observed in the pKO line.

We agree with the reviewer that this is a surprising and interesting observation. Our data suggest that the female position of the switch depends on the existence *in cis* of the genomic region covering intron1 and exons 2-4. Partial removal of this regulatory platform (i.e. removal of intron 1) shifts the sex ratio towards a male fate but still enables the production of females. The complete absence of intron 1 and exons 2-4 in $\Delta 270-699$ (pKO) precludes binding of the female switch activators causing the locus to be permanently active in the male position and resulting in an exclusively male population. In this model active RNA species produced in the female have a minimal role in the regulation itself which relies on activator binding - an interpretation that is supported by the absence of apparent phenotype in the females lacking *lncRNA* or *shortRNA* (KO).

Further understanding the architecture of the switch is complicated by the fact that modifying the coding sequence may have phenotypic outcomes which are the result of coding modification and not due to a change in the regulation of the locus. For instance, all our attempts to tag the N-terminus of the protein have reproduced the KO phenotype (i.e. females only). As a result, to be able to delve deeper into the architecture of the male switch, without editing the parasite's genome, we have used CRISPR interference and targeted dCas9 to the *mRNA* promoter. We found that this causes a shift towards more females (Extended data Fig. 6e,f), further supporting a model in which activators compete for the mutually exclusive access to the *md1* locus and consequently place it in one of the two states. We have enhanced the discussion around the possible regulation of the locus and its different states (Page 9, lines 16-34).

-The data presented may suggest that gametocytes develop as females by default, and only when *md1* is activated they develop as males. Is this the case, or female development requires the activation of alternative genes? Can this be established with the data available? This should be at least discussed.

Our data does suggest that in the absence of *md1* full length expression, gametocytes will develop as females, however we cannot rule out that additional factors are required in the female lineage.

-Sex determination and sexual conversion are separate processes. To avoid ambiguities, in the second line after the abstract, “sexual conversion” would be more appropriate than “sex”. In the first part of the abstract, “sex determination is epigenetic” is not strictly correct; sexual conversion has been demonstrated to be regulated by epigenetic mechanisms, but whether or not sex determination is also epigenetic is not known.

We have clarified both statements (Page 1, line 35 and Page 1, line 20)

-Md1 has not been described in any published article or annotated in PlasmoDB with this name. Therefore, the ID for this gene should be provided at the beginning of the article, not only in the Methods.

We have added the Gene ID in the first paragraph of the results (Page 2, line 2).

-In the first section of the results, the parasite stages used for the scRNAseq analysis should be described without having to go to the Methods. This is a fundamental information to interpret the scRNAseq figures. This is briefly described (“covering gametocyte maturation”) only in the next section describing the pKO line.

We have added this information in the first sentence describing the scRNAseq (Page 2, line 18 and Page 4, lines 22-23).

-To facilitate reading the manuscript, the two methods used for scRNAseq should be consistently named in the main text and Methods headings. A brief explanation of why different scRNAseq methods are used for different experiments should be provided.

Care has been taken to differentiate the two in the text/methods and an explanation has been added to the text (Page 8, line 43-45) and methods (Page 17, lines 41-47) to justify the use of either method to each line of experimentation.

-Fig. 1e (and similar panels in other figures). How were “progenitor” cells assigned to the female lineage or male lineage paths?

Two pseudotime developmental paths were computed starting from the same progenitor cells. The cells that were shared by both developmental paths were defined as progenitors whilst those that only belonged to one were associated with that sex (Page 19, lines 28-41).

-Are the genes in Fig. 1f potential Md1 targets? Do they share some common characteristics? This should be discussed.

The genes in Fig. 1f have diverse functions including functions associated with male development (e.g. p230p, heavy chain dyneins) but there is also a zinc finger protein which may be a further effector of male development. Our new data showed an association of Md1 with RNP granules - since we don't see an enrichment for this pathway in these differentially expressed genes, we don't think these would be direct targets of Md1. A reference highlighting genes previously associated with male development has been added to the legend of Fig. 1f (Page 3, lines 5-6).

-If I understand it correctly, the data for Male KO is from only 4 cells, whereas other columns probably include data from hundreds of cells. This should be explained in the figure legend.

The number of cells for each comparison (50 vs 50 for progenitors and 50 vs 4 for potential males) has been included in the legend of Fig. 1f (Page 4, lines 3-4).

-The schematic of Md1 domains (fig. 2e) should be presented earlier in the manuscript (after the second sentence in page 4). Please indicate the different cell types in fig. 2b.

Both requests have been completed (Page 4, lines 10-13; Fig. 2b).

-Fig. 3a and Ext. fig. 3b. Was a single read covering exon 1 sense identified? Is there an explanation for this? Even if the females are more abundant than males, the ratio of female-specific to male-specific transcripts appears to be unexpectedly high.

This is true, as mentioned above, we believe this is the result of (1) the female-biased sex ratio, (2) the relative higher number of copies of the *lncRNA* vs *mRNA* in females and males respectively and (3) a known bias of the method for shorter RNAs (PMID: 31740818).

-Fig. 3d. It is unclear how the primer spanning an intron was designed, given that multiple alternative splicing events appear to occur for the antisense. The position of the primers should be shown in the figure. Why were primers not designed against the region of the antisense transcript that does not overlap with the sense transcript?

Primers were designed to the dominant splice junction for the *lncRNA* and for the exon-exon borders for sense transcription. Primer positions have been added to Figure 3c (Page 8).

-Page 7. In the sentence "whereas following sex determination, the md1 locus can have two mutually exclusive bistable transcriptional states", following or preceding? If md1 is the regulator of sex determination, then its activation precedes sex determination.

The activation of the gene does indeed precede transcriptional sex determination, but we are unable to see in which state the locus is because directional mapping is less sensitive and require a higher expression level to be detected. We have amended the sentence to better reflect the likely underlying biology (Page 8, line 48)

-Fig. 4d is not cited in the text.

This has been corrected (Page 9, lines 29 and 34)

-The parasite line used (NF54) should be indicated in the first section of the Methods.

This has been corrected (Page 11, line 31)

-Page 13 bottom, the statement does not appear to refer to fig. 1f and 2e.

This reference has been removed.

-Ext. fig. 2b. Is it possible that the promoter driving expression of *hdhfr* has bidirectional activity, as reported for many malarial promoters, and this affects the regulation of the locus?

This is a possibility, the promoter driving *hdhfr* is in the reverse orientation to the *md1* gene (Extended data Fig. 2c) and could drive expression of the gene from the portion of exon 4 which is still present in the locus, however we do not see evidence for altered expression in this portion of the gene (Extended data Fig. 3c). Therefore, we don't believe this could affect the regulation of the locus.

Referee #2 (Remarks to the Author):

The mechanism for mating type determination in malaria parasites has been a topic of active investigation for over 40 years, and is extremely likely to be shared across malaria species, including *P. berghei* and *P. falciparum*.

An earlier study, with Dr. Talman as a lead author (ref 7), used a forward genetic knockout screen to identify 8 genes that are expressed during early gametocytogenesis of *P. berghei*, 4 of which are essential for male development specifically.

Of these, *Md1* was identified as the the most promising candidate for sex determination, based on its early expression following AP2-G induction.

Dr. Talman's team now examines the role of *Md1* in sex determination in *P. falciparum* in this manuscript,

Deletion of *Md1* had no effect on the asexual growth rate but following sexual commitment only female gametocytes developed, confirming the phenotype previously described in *P. berghei*. Complementation with full length *md1* restores the sex ratio to near wt.

Surprisingly, the generation of a *md1* truncation mutant that left the promoter and exon 1 intact resulted in a complete loss of female GCs and the remaining GCs only bifurcating into the male lineage.

However, maturation of these GC expressing male markers appears arrested based on both progression along the lineage and the inability to exflagellate.

Indeed, overexpressing the *md1* N-terminus in WT cells, greatly increased the fraction of male gametocytes, while overexpression of full length *md1* only skewed the ratio slightly.

Analysis of existing bulk RNAseq data and new direct RNA sequencing, identified 3 distinct transcripts from the *md1* locus, a sense transcript initiated upstream of exon 1 and both sense and anti-sense transcripts from a promoter inside the first intron.

While deletion of intron 1 completely abrogated anti-sense transcription resulted in a minor shift to more males (15% -> 20% males)

Fusion of a 2A-GFP reporter to *Md1* indicates that only the full length transcript is translated but the shorter transcript in females is not.

This study is elegant and carefully carried out.

Barring major issue #1 (see below), it marks a tremendous step forward in our understanding of sex determination in malaria parasites. In particular the observation that overexpression of the Md1 N-terminus can greatly increase males, shows that while the decision was likely made earlier, the mating type is not determined until one or the other isoform of Md1 is expressed.

MAJOR ISSUES:

1. Death of either male or female GCs in the KO or pKO lines would also result in a shift towards 100% of the other mating type.

The authors need to show that the total number of gametocytes formed in the pKO/KO lines remains the same in order to conclusively show that md1 determines the mating type rather than mediates survival of either males or females.

In other words, mutations in Md1 need to turn what would have been females into males (pKO) or vice versa (KO) rather than causing them to die or arrest.

We agree with the reviewer that this is a major point and we had addressed it in the initial manuscript showing the absence of aberrant morphological forms during development of the KO and appearance of some aberrant forms post determination of the $\Delta 270-699$ (pKO).

With the inclusion of new data showing that Md1 is translated from stage III gametocytes (between day 4-5), we would expect these aberrant forms to be present in our cultures from this stage onwards but make no such observation (Extended data Fig. 1d, Extended data Fig. 2h). Moreover, we would expect to detect aberrant or dying/dead cells in the single-cell experiments, but we see no evidence for cells in the KO or $\Delta 270-699$ (pKO) lines with a transcriptomic signature that is not also present in the WT (Fig. 1d, Fig. 2b). To ensure that we did not simply filter out poor quality cells (which would represent dead/aberrant), we have included QC metrics of the whole runs without filtering and haven't found cells in the KO absent in the WT (Extended data Fig. 1i). Although we do find a few $\Delta 270-699$ (pKO) cells of lower quality corresponding to the potentially degenerate developmentally-arrested male forms (Extended data Fig. 3b). Finally, in order to confirm this finding, we tracked the number of cells that retain a mitochondrial potential (using Mitotracker in flow cytometry) in a time course covering the sex determining event and show that following the drop in asexual parasitaemia (heparin preventing further invasion after day 2), WT, KO and $\Delta 270-699$ (pKO) population show a similar viability (Extended data Fig. 1e and Extended data Fig. 2f).

2. Using plaque assays Silvestrini and colleagues (PMID: 11128797) demonstrated quite convincingly that the mating type decision is already established in sexually committed schizonts formed at the end of the previous cycle.

Failure to reference this landmark paper deprives the reader of a crucial piece of context and needs to be thoroughly addressed in the context of the role of Md1.

Since Md1 isn't expressed until around day 3 of gametocytogenesis, (after AP2-G, which is expressed in committed schizonts and during the first 2 days of gametocytogenesis (PMID: 30478286)).

So while mating type isn't locked in until expression of the different isoforms of Md1 occurs, which isoform is going to be expressed is determined previously.

This is analogous to AP2-G, which depends upstream factors for its expression state but once that state is established the asexual/sexual differentiation is locked in.

We thank the reviewer for this comment. We have now included a thorough discussion of the timing of determination (Page 10, line 14- Page 11, line 10); including references to the papers from Silvestrini *et al.* and Smith and colleagues (PMID: 11085232), who do observe some mixed male and female plaques. We also include a discussion of the recent revisiting of commitment in both next cycle commitment and same cycle commitment (PMID: 30478286) and note that at least in same cycle commitment sex determination likely occurs later in development. Additionally, with the Silvestrini method it is also possible that plaques containing 2 or 3 gametocytes have matured in the same “sex-determining” micro-environment and may be therefore determined later in development whilst still displaying a consistent sex-bias. A further layer of uncertainty is also contributed by the demonstration of promiscuous expression of alpha tubulin 2 in both sexes (PMID: 21209927), which is the male-specific marker used by both Silvestrini and Smith. Altogether, although we do not discount the possibility of an earlier transcriptionally-silent determination of sex in schizonts, we believe that determination at the time Md1 starts being expressed is a likely scenario.

3. Gametocyte generation methods need to be specified in much greater detail. Since methods are not included in the word count there is no reason to keep them so brief.

This has been done (Page 11, line 36 - Page 12, line 20).

4. Calling all cells "progenitors" prior to bifurcation of the male/female lineages is unnecessarily vague, but the inability to distinguish them along the male/female axis isn't the only component that can be used to differentiate progenitors.

Many "Progenitors" are clearly early gametocytes, based on their expression of early gametocyte markers.

And since heparin was used to limit asexual growth, these should also include asexuals during at least the first 48 hours of gametocytogenesis, since heparin interferes with merozoite invasion.

At minimum, progenitors should be labeled as early gametocytes or trophozoites/schizonts to clear this up.

The stage composition of the progenitors should be specifically indicated and it needs to be clear to the reader that many progenitors are already gametocytes.

We agree with the reviewer that this is important, we have now specified the sexual progenitors and labelled the few asexual parasites that remain present in the datasets in the main figures (Fig. 1c and Extended data Fig. 1h for KO; Fig. 2b and Extended data Fig. 3a for $\Delta 270-699$ (pKO)). We have also added expression data of early sexual markers to highlight the population of early gametocytes in the datasets (Extended data Fig. 1h, Extended data Fig. 3a).

5. At what day of gametocytogenesis does md1 start being expressed? This can readily be determined using synchronous induction of gametocytes using the 2A-GFP line.

Using the suggested methods we have shown that Md1 starts being expressed at day 4-5 in stage III gametocytes, and have included this data (Extended data Fig. 6g,h).

6. The authors should comment on known functions of LOTUS/OST-HTH domains and how these might be involved driving the transcriptional programs underlying each mating type.

We have included this (Page 5, lines 2-17).

7. The fact that nearly all GCs are male when Md1 is truncated after exon 1 but only 20% are male when full length md1 is expressed suggest that the region of Md1 protein encoded by exons 2-5 negatively regulates the region of Md1 encoded

We agree with the reviewer that the region encoding *md1* exon 2-4 does indeed seem to be required to regulate expression of the full-length transcript. We don't believe there is regulation at the level of the protein since it is only translated in males. The discussion around the regulation of the locus has been enhanced (Page 9, lines 16-34).

8. The authors show md1 is essential for sex-determination and that a bistable switch exists that leads to two distinct transcriptional states at the md1 locus.

However, that doesn't mean that md1 is the bistable switch or part of the bistable switch.

Indeed, the fact that deletion of intron 1 had only minimal effects on sex ratio indicates that md1 is downstream of the switch.

That said, the fact that Md1 truncation or complete deletion switches the sex ratio one way or another shows that this decision manifests and locks in only upon expression of md1.

We believe exon 2-4 retains some activity of the switch even in the absence of intron1 and *lncRNA*; our data indicate that *md1* is in two states that are linked to sex determination however we do agree that it may not be the whole or definitive part of the switch (Page 9, lines 16-34).

MINOR ISSUES:

Use of pKO for partial KO is confusing since it looks like a plasmid name. a more descriptive name like (like md1 Δ 270-699) should be used.

The name has been changed to Δ 270-699 (pKO).

Fig1C which DHC? There are several with sex specific expression.

The geneID (PF3D7_0905300) has been added to the legends (Fig. 1c, Fig. 2b, Fig. 4a).

Fig S1e,S2d,S2k: dont normalize to female marker (over-estimates the change since they aren't independent but anti-correlated, use uce instead)

This has been changed (Extended data Fig. 1f, Extended data Fig. 2e, Extended data Fig. 3f).

Fig S2i: indicated the deleted regions for pKO and KO

This has been added (Page 26, Extended data Fig. 3c).

How does the male trajectory differ in pKO?

This male trajectory in the Δ 270-699 (pKO) is characterised by an early arrest in the male lineage. The expression of *md1* is still silenced prior to determination, and switched on with a timing similar to the WT, suggesting the regulation of initial expression is determined by the promoter region.

RNAs: what are the approximate 5' & 3' UTRs for md1-mRNA & md1-ssRNA ? are md1-ssRNA or md1-lncRNA polyadenylated?

Annotated 5' & 3' UTRs for md1-mRNA are 643 and 584 bp respectively. We do believe shortRNA to be polyadenylated based on our sequencing data, on the other hand given the extreme AT content of the *lncRNA* it is difficult to make this determination with certainty.

Please chose another name for the exon 2-4 sense transcript. md1-ssRNA looks like single-stranded RNA not short-sense.

We have amended the nomenclature to shortRNA (Fig. 3).

Fig. S3a: reads from male vs female Lasonder is from much later in GC development

Yes, this has been highlighted in the legend (Page 28, Extended data Fig. 5a).

Fig. S3b: sense reads from exon 1 are much lower.

This is true, as mentioned above, we believe this is the result of (1) the female-biased sex ratio, (2) the relative higher number of copies of the *lncRNA* vs *mRNA* in females and males respectively and (3) a known bias for the method for shorter RNAs (PMID: 31740818).

Fig. 3c show positions of primers used in 3d.

Primer positions have been added (Page 8, Fig. 3c).

Fig. 3c The relative abundance of the full length vs ssRNA transcripts should also be shown. (qRT-PCR with primers in exons 2-5 (ssRNA+mRNA) vs exon1 (mRNA) with sense specific primers used for RT)

This has been added in the Extended data Fig. 5e,f (Page 28) as well as primer positions (Page 8, Fig. 3c).

Δ intron1: "promoter of md1-lncRNA spans intron 1" cant infer length of the promoter but instead refer to speak to TSS

We now refer to TSS (Page 7, line 23)

What are the cells shown in Figure 4?

Maturing gametocytes of the Md1-2A-GFP, this has been added to the legend (Page 10, lines 4-5).

Plot ap2-g expression on cell fate plots
plot pfs16 on male female

This has been added (Extended data Fig. 1h, Extended data Fig. 3a).

4B middle panel doesn't support statement "in the male lineage, the md1 promoter first becomes active before cells can be unambiguously identified as males and is coupled with protein expression." Just because you can't distinguish their fate transcriptionally doesn't mean that the decision has not already been made but hasn't resulted in transcriptional differences.

The statement has been amended to “unambiguously identified on a transcriptional level” (Page 9, lines 8-9).

Referee #3 (Remarks to the Author):

In malaria parasites, sexual reproduction coincides with transmission to the vector host, and therefore the understanding of the mechanisms of sex determination and differentiation in *Plasmodium falciparum* are object of intense research. Sex in these organisms is determined epigenetically, i.e., male and females are genetically identical. The events that regulate the differentiation of sexually committed parasites into either male or female pathways are not well understood, and this paper represents a significant advance in this direction. In particular, the manuscript suggests that Md1 is the key gene in determining the gender of *Plasmodium falciparum*, and proposes an association of the male-determining function with the N-terminus of the Md1 protein, whereas the C-terminus may be involved in male developmental regulation. Moreover, it is suggested that sex determination involves a bistable genetic switch at the Md1 locus where an alternative version of Md1 would produce a lncRNA that ensures silencing of the male fate in female cells.

The manuscript is overall well written and well presented, and tackles an important and old question, both in an applied and fundamental perspective. The approaches used are valid, and data is of high quality. The conclusions are sound, but I have a number of concerns and request for clarifications, described below point by point.

1) How does this manuscript link to a Bioarchives preprint that also describes md1 as one of the ten genes involved in sexual determination/differentiation in *P. berghei*? The main author of the manuscript is one of the first authors of this manuscript. The pre-print describes a screen for genes required for the formation of male and female gametocytes in *P. berghei*. The preprint also uses single-cell transcriptomics to map the sexual differentiation events of lineages at high temporal resolution. Md1 is identified as (one of the) male-determining gene and highlights the domain LOTUS/OST-HT. This preprint describes md1 but also identifies a second locus, md2, as potential male-determining, since disruption leads to a complete loss of cells expressing male markers. In particular, md1 is shown to be the most noticeable AP2-G responsive gene to be upregulated early in the sexual pathway (which I guess was the reason why the authors chose to work on md1 in the present manuscript?).

Therefore, there is (at least conceptual and technical) some overlap with the current paper, and it would be important to clarify how the two papers articulate.

As noted above, the bioRxiv manuscript, a primary screen for sexual development in the rodent malaria model, identifies possible regulators but does not establish at what point in development the different candidate genes may act. As such the bioRxiv manuscript doesn't probe sex determination but rather sexual development as a whole. It is also conducted in the less relevant and more experimentally tractable rodent model. Importantly, the current paper whilst it builds on the identification of Md1 from our previous efforts, is the first to identify a central regulator of sex determination in *Plasmodium* and more generally in Apicomplexa.

2) The gene md2 (for which no detailed information is given in the preprint except that it is not conserved outside *Plasmodium*) appears to have a role in sex (male) determination and I wonder how these findings fit in the model proposed by the authors that md1 is necessary and sufficient to induce a male cell fate. For example, key information could be obtained if some of the experiments of the current paper were performed in a md2 background.

Whilst *Pbmd2* has a possible ortholog in *P. falciparum* (PF3D7_1233200), the two proteins only share 60% identity covering a small portion of the protein (7% coverage) and we could not detect PF3D7_1233200 expression in any of our datasets covering sex determination, as opposed to a strong expression in *P. berghei* with the same methods. We, therefore, believe it is unlikely that PF3D7_1233200 has a conserved role in sex determination. This further stresses the importance of conducting studies in the human malaria species. The long-term goal of identifying other players in the pathway is ongoing and will be the subject of future manuscripts. We also note that repeating our experiments in another genetic background would be an immense undertaking and quite uncertain given the difficulty of doing several rounds of genetic modifications whilst preserving the ability of the parasite to differentiate sexually.

3) I have some reservations regarding the way phenotypic sex was determined, because as far as I understood, markers have been used to infer sex of the gametocytes, but not functional sex (crosses with opposite sex). Because sex is not genetic, these markers correspond to 'sex-biased genes' (that have been previously identified). Without a clear determination of the phenotypic sex of the gametocytes, using genetic crosses which reveal the 'real' functional sex, the whole interpretation of the results may be flawed, biased and/or incomplete. This is because the cascade of genes involved in male and female development ('sex-biased genes') may be decoupled with the 'functional' sex (for example, lines may be expressing female markers – i.e., be feminized – but still be functionally males, in the sense they fuse with female gametes). For example, can you exclude that the KO is simply affected in some parts of the sex-differentiation cascade (i.e., the line may be 'feminized' based on the markers), but still be phenotypically male (i.e., would be capable of fusion with female gametes)? Controlled genetic crosses appear to be feasible in Plasmodium because I see they were performed in the preprint (Russel et al).

This is an astute point, and indeed our marker-based determination of sex using just two sex-specific markers (the protein marker Pfg377 and the mRNA marker PfmGET) would be susceptible to this concern. However, we feel that the use of single cell transcriptomes precludes this possibility, as the male and female expression programs are different by several hundred expressed genes and we have confidently identified males (WT and $\Delta 270-699$ (pKO)) or have found canonical females (WT and KO) in the datasets (Fig. 1d, Fig. 2c). For the KO we observe no difference between WT and KO females, indicating that the females present in the KO have the full wild-type female gene expression signature rather than males expressing just a few female markers. The same holds for the $\Delta 270-699$ (pKO) parasites - although the males are arrested at an earlier stage of development they still display a developing male phenotype. Further supporting this notion is the absence of all intermediate developmental male stages in the KO and female stages in $\Delta 270-699$ (pKO) signifying that the entire developmental program is absent. We also test male gametogenesis (i.e. the cascade of 3 rounds of DNA replication followed by male flagellar production leading to a 'sperm' like gamete) which is the functional outcome of successful male gametocytogenesis, which supports this conclusion. We therefore believe crosses are not needed to support our conclusions and also note that they would be technically very challenging in *P. falciparum*. These require BSL3 facilities and are only routinely performed in the more experimentally and genetically-tractable rodent model.

4) Did I understand correctly that two mutants were produced? KO (full) and pKO (partial)? But only one full KO? Considering the effect of KO and pKP is different in terms of phenotype, wouldn't you need a couple of independent mutant alleles to validate the link between phenotype and genotype? Or do you consider that KO and pKO are 'alleles'?

We don't consider these modifications alleles. We have edited the haploid parasite with CRISPR-Cas9. We provided DNA templates that allow the repair of the double strand breaks via double homologous recombination (the dominant repair pathway in *Plasmodium*). The locus is thus modified in exactly the intended way and in each case, the correct expected genome modification has been validated. We had initially generated $\Delta 270-699$ (pKO) and found a surprising and fortuitous phenotype, due to the partial expression of the remaining truncated form as is demonstrated in the manuscript. This warranted the complete removal of the locus which was more difficult to implement because of the position of available gRNAs in the highly AT-rich genome. For each modification we have characterised two different cloned mutants of the parasites (Extended data Fig. 1b, Extended data Fig. 2d). As a further genetic validation step, we have also complemented the phenotype of the KO line with episomal expression of the full-length *md1* gene, which complemented the phenotype (Fig. 1b).

5) If Md1 is involved in sex determination, one would expect that the protein Md1 should be produced in males gametocytes (but not female)? The manuscript is missing *md1* protein analysis., using a proteomic dataset would allow to verify that *md1* is only present in male gametocytes and provide additional evidence that this gene is indeed exclusive to the male pathway. Perhaps sex-specific proteomic datasets are already available and could be explored.

This is a correct assumption, and an existing proteomic dataset (PMID: 27298255) reveals that Md1 is enriched in a male-enriched fraction. Furthermore, the translational reporter line (Md1-2A-GFP) demonstrates the protein is generated just after males can be transcriptionally identified and is specific to the male lineage consistent with our hypothesis (Fig. 4b, right panel).

6) 'Timing' of male fate and expression of the *md1* needs clarification. The authors state that *md1* mRNA is present after the fate is determined "exon 1-2 junction (which can only arise from *md1*-mRNA) was exclusively expressed immediately following bifurcation into the male branch". However, if *md1* is involved in sex determination, it would be expected that the transcript (and protein) should be present before the bifurcation already, in the cells that will become male, and not once the fate has been already determined. Indeed, it is stated later in the text "In the male lineage, the *md1* promoter first becomes active before cells can be unambiguously identified as males and is coupled with protein expression." however, I don't see where is the figure/results that support this statement, perhaps I missed it?

We have now amended the text to clarify this point (Page 8, lines 48-53). Fig. 4c and Extended Data Fig. 6c,d show that *md1* is first detected before the two lineages branch out but at this point we do not detect stranded counts (which are less sensitive to detection because they require exon-exon split reads); this only becomes detectable as cells progress in either lineage.

7) the authors remove intron1 (Δ int1 line) and confirm that the intron is indeed important for *md1*-lncRNA transcription. I could not find the control showing that *md1* is expressed to normal levels (and that the *md1* protein is there – see my comment below about proteomics) in the Δ int1 line?

We thank the reviewer for this suggestion, we have added a qPCR showing expression of *md1*-mRNA (at exon 1-2 and exon 2-3 junctions) (Extended data Fig. e,f). We have also now characterised WT, Δ int1 and $\Delta 270-699$ (pKO) by shotgun proteomics and verified that the protein is indeed expressed in Δ int1 (Extended data Fig. 5g), we also find that the $\Delta 270-699$ (pKO) expresses Md1 with peptides covering the N-terminus only as expected (Extended data Fig. 3e).

8) I may be confused here, but if *Md1* is already differentially expressed in the progenitor cells (upregulated ? see fig 1f), how is it possible that a proportion of the progenitor cells will become female? Does this mean there are genes involved in the female fate that override *md1* in these cells? Again, knowledge about the presence of the *md1* protein in progenitor vs male and female cells would be perhaps useful. Importantly, the authors cannot exclude that other genes are involved in the pathway. As indicated in the preprint, several other genes are identified as related to sex determination, so I would be careful in not overstating *Md1* as necessary and sufficient for sex determination, it is probably one (key) actor in the pathway.

This is correct, *md1* is expressed in progenitor cells in the 10X dataset (Extended data Fig. 6d) and we can also see it already expressed in some progenitor cells in our smartseq2 dataset (Extended data Fig. 6c). However this corresponds to non-stranded counts and therefore could be either from sense or antisense transcription. Although we mapped the directionality of transcription at the *md1* locus, this requires exon-exon spanning reads and is less sensitive than simple mapping to the locus. We therefore believe that the locus is already in its male or female state in the progenitor but that the orientation of transcription is not yet apparent.

We entirely agree with the reviewer that other genes are likely involved in this process and have further taken this into account in our discussion of the sex determining pathway (Page 9, line 16-34).

9) The authors state that “*md1*-lncRNA is not the main driver of transcriptional silencing of the *md1* promoter in females.” Do you have an idea of what is the target of the *md* lncRNA in females? The authors have several datasets on genes involved in the sex determination pathways, could any of the genes that are silenced in females (strongly upregulated in males) be the target?

That’s a good question. Surprisingly females from the KO and WT have no differentially expressed genes. Since the *lncRNA* is absent in KO females, it would suggest a minimal role of the *lncRNA*.

10) I find a bit confusing that a range of different male and female markers are used throughout...is there a way to simplify this? Perhaps listing in the methods clearly which are the male and female markers. Actually, I was wondering why different markers are used for different experiments? I would have thought it is better to have several male and female markers per experiment (instead of only one marker per experiment, and different markers in different experiments). What was the basis of the choice of the different markers in each experiment? Also, I would like to have some clarification concerning the use of PG377 as a female marker, as this marker, if I understand right, only works for late stages of gametogenesis. Could the fact that this marker does not capture female fate in early stages affect your results?

We have chosen to use the two published sex determination methods. One using Pfg377 protein (PMID: 21209927) and one using PfmGET mRNA (PMID:30497508), since they are well validated in the literature and probe different aspects of cellular sex specificity (osmiophilic body protein, or mRNA marker). The reviewer is correct that Pfg377 is only validated as a stage V marker and that is when we performed the phenotyping for these two assays. The reviewer is correct that this single-marker analysis alone could miss an “arrested/dead” form confounding the results, however the time course and single cell data analysis throughout gametocytogenesis preclude this possibility. For the single cell experiments we have used our previously established markers (PMID: 31439762) to illustrate the different dimensionality reduction and orient the reader.

Finally, the question that is not answered and that would be needed to fully apprehend the sex determining pathway is what actually triggers the switch (what triggers the production of the sense

transcript 'md1-mRNA' in males and lncRNA transcript in females). Perhaps one of the genes that are described in the preprint? Although I am not requesting the authors to do this, but I think it is important to emphasize that Md1 is involved in sex determination, but may not be the one and only 'master' sex-determining gene.

This point is especially true and important, as sex determination is thought to be environmentally-responsive and the cell fate decision pathway is likely to include components that regulate effectors such as Md1. Going "back up" the differentiation pathway is especially challenging and requires single loci proteomics approaches. We have incorporated a more detailed discussion including the placement of Md1 in the overall cell fate decision (Page 9, line 16-34).

Other comments:

Figure 1c) what is the different between the upper and lower panels?

The upper panel gives the expression of genes that are known to be male or female, the lower panel displays the computed pseudotime ordering in development which is further used in e. We altered the legend to clarify this point (Page 3, lines 10-11).

Figure 1f) shows several DE genes there, among them there is md1. What are these other genes? Are they also involved in sex determination?

The other genes show diverse functions including functions associated with male development (e.g. p230p, heavy chain dyneins) but there is also a zinc finger protein which may be a further effector of male development. Further study will be needed to determine if these genes are in the male differentiation regulatory pathway. Known genes have been highlighted (Page 4, lines 5-6).

Section "Md1 is necessary to determine a male fate". If I understood correctly, there is a reduction of proportion of mature males, however, if this gene is the male master sex determinant, then the effect of the KO should be seen early too. Is this the case?

This is correct. Our data points to a complete absence of males in the KO. Using the cruder Pfg377 phenotypic assay we see a dramatic increase in the number of Pfg377+ positive cells (females), with a few remaining negative (which could be male or the natural proportion of unstained or dead cells). However, using single cell transcriptomics we see no bifurcation in the male branch from the earliest point of male maturation and observe a total absence of males showing the necessity of Md1 to engage in the male pathway from the earliest point of differentiation (Fig. 1d).

Extended fig 1e –In order to obtain a mean logFC of about 0.5 in the WT (or at least a positive value) you would need to have more male than female marker (because normalization is performed based on male/female marker?). If this is the case, there is a strong male bias in the WT culture. However, in the text it is stated that there are about 85% female (Pfg377)-positive gametocytes in the wild type (because there is a strong female-bias). Isn't there an inconsistency somewhere, or perhaps I am misunderstanding?

The relative fold change was calculated with the delta delta Ct method with the WT as a control and the genes likely have different copy numbers per cell. As suggested below the normalisation has now been amended to use a housekeeping gene (*uce*) (Extended data Fig. 1f, Extended data Fig. 2e, Extended data Fig. 3f).

It would have been useful to have northern blots to validate the splice variants of md1 ?

We provided a long read direct RNAseq (ONT) dataset showing the full-length species of variants from *md1* and validated them using splicing analysis in short read sequencing (Illumina). We are unsure of how much more information/validation a Northern blot would provide.

Related to this, I don't understand why the authors don't normalize the QPCR using a normalization gene (i.e. that has a constant expression, regardless of the sex). This needs to be clarified.

As suggested, we have now normalised qPCR with the housekeeping gene *uce* (PF3D7_0812600) (Extended data Fig. 1f, Extended data Fig. 2e, Extended data Fig. 3f).

Extended fig 1d – for consistency, please present data also for KO G10.

We have included this data (Page 24, Extended data Fig. 1d).

“To avoid the pitfalls of distinguishing gametocyte sex morphologically”

Statistical analysis to support the sentence “The proportion of Pfg377-positive gametocytes was 85% in the WT (fig1b), indicating a typical female-biased sex ratio. In contrast, KO gametocytes were almost exclusively Pfg377-positive (97%), indicating very few, if any, males in the population.” is required. In the legend of the figure it is mentioned that “n=4” but no indication of numbers of cells nor the statistical tests that were performed. Is 85% vs 97% significantly different? What does the n=4 mean in the legend? Do I understand correctly that you used 4 different mutant lines (independently mutated/alleles)? How many gametocytes were counted in each line? What do the asterisks correspond to? Detailed stats explanations, including what you call ‘n’ are also missing in legends for example Extended figure 2k, 2d, 2e, etc.

We have added the required information in all figure legends including number of biological replicates and statistical test used, the number of cells counted is given in the methods (Page 15, line 14) and the statistical analysis is further described in the method (Page 20, line 22-30).

“Taken together, this suggests that *md1*-lncRNA is dispensable for male determination and its knockdown may even enhance bifurcation to a male fate.” – How do you explain this observation that there is even an increase in male fate in absence of *md1*-lncRNA?

The most parsimonious explanation of our data is that the *lncRNA* is a result of the switch in the “female” position and not itself an effector. In the KO, where the *lncRNA* is completely absent, we see completely canonical females suggesting that the *lncRNA* is dispensable for female development. In the Δ int1 line, we see that the switch can still be in the “female” position even in the absence of intron 1 (and the *lncRNA*). We interpret the slight increase in males in the Δ int1 to be a consequence of a decrease in accessibility of activators to the female switch (only reduced without the intron but still somewhat functional in the unperturbed exon2-4 region) that leads to an increase in likelihood of having the switch on “male” position.

Perhaps a more philosophical question is why we use the term ‘sexes’ and not ‘mating type’ – given that these organisms appear not to be clearly anisogamous/oogamous (eggs and sperm), so no asymmetry between gametes?

When gametocytes reach the mosquito they further differentiate into dimorphic gametes (egg and sperm-like) explaining the anisogamous nomenclature.

Notes

- The supplementary tables containing 10X count matrices have been enriched to include metadata as requested during the initial review by one of the reviewers.
- Point number 3 was missing from reviewer 3's list of comments, we have renumbered the list to avoid any omission on our part.
- Given the number of new data panels, 2 new Extended data figures have been added.

Reviewer Reports on the First Revision:

Referees' comments:

Referee #1 (Remarks to the Author):

The revised manuscript by Gomes et al has improved substantially and the response to the majority of my comments is satisfactory. The localization of Md1 in the cytoplasm and the description of its interactome are important additions that provide insight into the molecular function of the protein.

There are two aspects of the manuscript on which I am still not convinced:

1-The authors conclude that sexual determination occurs at stage III of sexual development, based on experiments with a parasite line in which a GFP tag is appended to Md1. I think this conclusion is not demonstrated, and therefore some statements should be toned down (e.g. page 11, line 10). The new data with the GFP line can of course be presented, but it is possible that expression of the protein starts later than the actual sex determination event. It is also possible that sex is determined earlier than the male-female branching point in the scRNAseq PCA, but the algorithm cannot confidently classify these cells as males or females because differences occur in the expression of only a very small subset of genes.

Additionally, data in Extended fig. 1h, 3a and 6c suggest that pfap2-g is still highly expressed near the male-female branching point. Expression of this gene has been shown to be largely reduced before stage III. Furthermore, the expression of early gametocyte markers such as pfs16 and gexp05 appears to start near the branching point, and the onset of expression of these genes is well-known to start in stage I gametocytes or even earlier. Therefore, I have doubts about the conclusion that the branching point occurs at stage III.

Would it be possible to map the developmental stage of parasites at specific pseudotimes (e.g., cells near the branching point) to specific gametocyte development stages by comparing their expression patterns with high resolution RNA-seq datasets across gametocyte development?

2-The model for the transcriptional switch at the md1 locus is not convincing and should be revised.

What are the md1 transcripts detected before bifurcation, if neither md1-mRNA or md1-lncRNA are detected before the bifurcation? The statement "The md1 promoter first becomes active before cells can be identified as belonging to one sex or the other" (page 9 line 6) is not supported by the data, because activation of this promoter would lead to md1-mRNA transcripts, and they are not detected before bifurcation. Perhaps the authors referred to the intron promoter, rather than to the md1 promoter? Furthermore, the schematic in fig. 4d shows no transcripts in progenitor cells, which is inconsistent with the data presented and with the statement above (non-stranded md1 reads are detected in progenitor cells).

The proposed mutually exclusive binding of specific regulators, involving competition between the md1 promoter and a site located between intron 1 and exon 4, is in my opinion too speculative

(especially the putative binding of the factors to exons 2-4). Rather than competition between broad DNA regions for binding by a hypothetical factor, the data suggests that chromosome conformation, TADs or long-range interactions may be involved, and deletion of specific parts of the locus may affect differentially a delicate equilibrium. The model should be presented more conservatively and mentioning alternative possible explanations. The authors made an important effort to support their model with the dCas9 experiments but the differences observed were modest and the result is clearly open to alternative interpretations.

Referee #2 (Remarks to the Author):

The authors have carefully and comprehensively addressed my concerns. Outstanding work!

Referee #3 (Remarks to the Author):

The authors have thoroughly answered to all my comments and concerns. Although I still believe it would have been useful to test the phenotypic sex of the presumably sex reversed knockouts, I understand this is technically very challenging. I would like the authors to consider adding a sentence about this caveat in the manuscript.

Author Rebuttals to First Revision:

Referees' comments:

Referee #1 (Remarks to the Author):

The revised manuscript by Gomes et al has improved substantially and the response to the majority of my comments is satisfactory. The localization of Md1 in the cytoplasm and the description of its interactome are important additions that provide insight into the molecular function of the protein.

There are two aspects of the manuscript on which I am still not convinced:

1-The authors conclude that sexual determination occurs at stage III of sexual development, based on experiments with a parasite line in which a GFP tag is appended to Md1. I think this conclusion is not demonstrated, and therefore some statements should be toned down (e.g. page 11, line 10). The new data with the GFP line can of course be presented, but it is possible that expression of the protein starts later than the actual sex determination event. It is also possible that sex is determined earlier than the male-female branching point in the scRNAseq PCA, but the algorithm cannot confidently classify these cells as males or females because differences occur in the expression of only a very small subset of genes.

Additionally, data in Extended fig. 1h, 3a and 6c suggest that pfap2-g is still highly expressed near the male-female branching point. Expression of this gene has been shown to be largely reduced before stage III. Furthermore, the expression of early gametocyte markers such as pfs16 and gexp05 appears to start near the branching point, and the onset of expression of these genes is well-known to start in stage I gametocytes or even earlier. Therefore, I have doubts about the conclusion that the branching point occurs at stage III.

Would it be possible to map the developmental stage of parasites at specific pseudotimes (e.g., cells near the branching point) to specific gametocyte development stages by comparing their expression patterns with high resolution RNA-seq datasets across gametocyte development?

We agree with the reviewer, that detection by microscopy of Md1-GFP may not be sensitive enough to detect when Md1 is first expressed and therefore the branching point may precede Md1-GFP detection. We have conducted a correlation analysis of our scRNAseq dataset with a published high-resolution bulk dataset as suggested (van Biljon et al., 2019. doi:10.1186/s12864-019-6322-9). Result are displayed here:

Whilst all stages prior to the branching event most correlate to Stage II gametocyte, during the branching event cells map to stage II/stage II-III/StageIII-IV. This does indeed suggest that

determination may precede full maturation into stage III and occurs during the transition from stage II to stage III, the manuscript has been amended to reflect this and to caution against the possible lag between actual sex determination and Md1-GFP protein detection. “Although there may be a lag between sex determination and appearance of GFP, our data strongly suggest sex determination occurs around day 3-4 of gametocytogenesis during the stage II-III transition”)

2-The model for the transcriptional switch at the *md1* locus is not convincing and should be revised.

What are the *md1* transcripts detected before bifurcation, if neither *md1*-mRNA or *md1*-lncRNA are detected before the bifurcation? The statement “The *md1* promoter first becomes active before cells can be identified as belonging to one sex or the other” (page 9 line 6) is not supported by the data, because activation of this promoter would lead to *md1*-mRNA transcripts, and they are not detected before bifurcation. Perhaps the authors referred to the intron promoter, rather than to the *md1* promoter? Furthermore, the schematic in fig. 4d shows no transcripts in progenitor cells, which is inconsistent with the data presented and with the statement above (non-stranded *md1* reads are detected in progenitor cells).

The statement is indeed potentially misleading and has been amended to “The *md1* locus first becomes active before cells can be identified as belonging to one sex or the other”. *Md1* reads identified in progenitors are in the body of the gene and do not span introns so we cannot attribute them to a specific *md1* RNA species. Split reads are rarer and their detection less sensitive, corresponding to a lower expression in progenitors.

The schematic was intended to indicate the underlying cellular state before the locus becomes active, there is necessarily a lag between *md1* expression in either male or female modes and observable transcriptional consequences. We hypothesise that the sexual progenitor’s expression corresponds to cells already committed but that cannot be sexed by scRNAseq yet. The legend has been amended to reflect this.

The proposed mutually exclusive binding of specific regulators, involving competition between the *md1* promoter and a site located between intron 1 and exon 4, is in my opinion too speculative (especially the putative binding of the factors to exons 2-4). Rather than competition between broad DNA regions for binding by a hypothetical factor, the data suggests that chromosome conformation, TADs or long-range interactions may be involved, and deletion of specific parts of the locus may affect differentially a delicate equilibrium. The model should be presented more conservatively and mentioning alternative possible explanations. The authors made an important effort to support their model with the dCas9 experiments but the differences observed were modest and the result is clearly open to alternative interpretations.

We agree with the reviewer that alternative interpretations are possible and have amended the text to reflect these. (“The two transcriptional states could therefore be the result of differential chromosome conformation or compartmentalization, or mutually-exclusive binding of specific regulators either to the promoter or to intron 1 through exon 4”)

Referee #2 (Remarks to the Author):

The authors have carefully and comprehensively addressed my concerns. Outstanding work!

We thank the reviewer for their appreciation of the work.

Referee #3 (Remarks to the Author):

The authors have thoroughly answered to all my comments and concerns. Although I still believe it would have been useful to test the phenotypic sex of the presumably sex reversed knockouts, I understand this is technically very challenging. I would like the authors to consider adding a sentence about this caveat in the manuscript.

We thank the reviewer for their appreciation of the incorporated modifications. Although we agree that performing crosses is a valid approach, we found it difficult to incorporate a statement in the manuscript without altering the narrative flow while simultaneously removing significant amounts of text due to length constraints.

Reviewer Reports on the Second Revision:

Referees' comments:

Referee #1 (Remarks to the Author):

My suggestions have been addressed and the conclusions that were in my opinion less convincing are now interpreted more cautiously.

The correlation analysis with the van Biljon dataset is interesting. Although it is not conclusive, the authors may wish to include this analysis in a supplementary figure to support the view that sex determination likely occurs much later than previously proposed and generally accepted. However, the almost complete absence of cells assigned to stages I or I-II may be a consequence of mixture of sexual and asexual parasite at the early time points in the van Biljon dataset (which should be acknowledged if the figure is presented).

I congratulate the authors for an excellent article that provides a major advance in malaria parasite biology.